# GWAS meta-analysis of intrahepatic cholestasis of pregnancy implicates multiple hepatic genes and regulatory elements

Peter H. Dixon [1,176], Adam P. Levine [2,3,176], Inês Cebola [4,176], Melanie M. Y. Chan [2], Aliya S. Amin[1], Anshul Aich[2], Monika Mozere[2], Hannah Maude[4], Alice L. Mitchell[1], Jun Zhang[2,5], NIHR BioResource[6*], Genomics England Research Consortium Collaborators[7*], Jenny Chambers[8,9], Argyro Syngelaki[10], Jennifer Donnelly[11], Sharon Cooley[11], Michael Geary[11], Kypros Nicolaides[10], Malin Thorsell[12], William M. Hague[13], Maria Cecilia Estiu[14], Hanns-Ulrich Marschall [15], Daniel P. Gale [2,176] & Catherine Williamson [1,176✉]

Intrahepatic cholestasis of pregnancy (ICP) is a pregnancy-specific liver disorder affecting 0.5–2% of pregnancies. The majority of cases present in the third trimester with pruritus, elevated serum bile acids and abnormal serum liver tests. ICP is associated with an increased risk of adverse outcomes, including spontaneous preterm birth and stillbirth. Whilst rare mutations affecting hepatobiliary transporters contribute to the aetiology of ICP, the role of common genetic variation in ICP has not been systematically characterised to date. Here, we perform genome-wide association studies (GWAS) and meta-analyses for ICP across three studies including 1138 cases and 153,642 controls. Eleven loci achieve genome-wide significance and have been further investigated and fine-mapped using functional genomics approaches. Our results pinpoint common sequence variation in liver-enriched genes and liver-specific *cis*-regulatory elements as contributing mechanisms to ICP susceptibility.

[1] Department of Women and Children's Health, School of Life Course Sciences, King's College London, London, UK. [2] Department of Renal Medicine, University College London, London, UK. [3] Research Department of Pathology, University College London, London, UK. [4] Section of Genetics and Genomics, Department of Metabolism, Digestion and Reproduction, Imperial College London, London, UK. [5] Division of Nephrology, Department of Medicine, Third Affiliated Hospital of Sun Yat-Sen University, Guangzhou, Guangdong, China. [6] NIHR BioResource, Cambridge University Hospitals, Cambridge Biomedical Campus, Cambridge, UK. [7] Genomics England, Queen Mary University of London, London, UK. [8] ICP Support, 69 Mere Green Road, Sutton Coldfield, UK. [9] Women's Health Research Centre, Imperial College London, London, UK. [10] Harris Birthright Research Centre for Fetal Medicine, King's College Hospital, London, UK. [11] The Rotunda Hospital, Dublin, Ireland. [12] BB Stockholm, Danderyd Hospital, Stockholm, Sweden. [13] Robinson Research Institute, The University of Adelaide, Adelaide, SA, Australia. [14] Ramón Sardá Mother's and Children's Hospital, Buenos Aires, Argentina. [15] Department of Molecular and Clinical Medicine/Wallenberg Laboratory, University of Gothenburg, Gothenburg, Sweden. [176]These authors contributed equally: Peter H. Dixon, Adam P. Levine, Inês Cebola, Daniel P. Gale, Catherine Williamson. *Lists of authors and their affiliations appear at the end of the paper. ✉email: catherine.williamson@kcl.ac.uk

ntrahepatic cholestasis of pregnancy (ICP) is a pregnancy-specific liver disorder affecting ~0.5–2% of pregnant women. The majority of cases present in the third trimester with pruritus. The diagnosis is confirmed by elevated maternal total serum bile acids (TSBA) and abnormal serum liver tests[1]. Elevation of maternal TSBA ≥40 µmol/L is associated with an increased incidence of adverse pregnancy outcomes[2,3], and a large meta-analysis of ICP cases identified 100 µmol/L as the threshold at which the risk of stillbirth increases[4].

ICP has a complex etiology, with genetic, endocrine, and environmental contributions. A variable geographical prevalence has also been observed with this disease. This may be explained by genetic variation in women of different ancestry, or could be the result of altered environmental factors, e.g., vitamin D or selenium deficiency[5]. The cholestatic effect of gestational elevations of estrogen and progesterone metabolites unmasks the disease in genetically susceptible individuals[6,7]. Rare mutations in the hepatobiliary transporter genes *ABCB4* and *ABCB11* (encoding the phosphatidyl choline floppase and the bile salt export pump, respectively) have been reported, with the most recent study identifying pathogenic or likely pathogenic heterozygous mutations in either of these two genes in up to 25% of cases of severe, early-onset ICP[8]. A limited role has been proposed for rare genetic variants in other known cholestatic loci[9]. In contrast with rare variants, the contribution of common genetic variation to the heritability of ICP has not been systematically addressed thus far. Candidate gene analyses have been reported[10] but to date the only genome-wide study of ICP has employed admixture mapping of genetically distinct populations[11]. Here, we utilized the recent whole-genome sequencing study by the NIHR BioResource Rare Disease collaboration (NIHR-RD) and the 100,000 Genomes Project (100KGP)[12] along with data from the FinnGen Consortium to systematically identify common variants that contribute to ICP susceptibility. We have performed the first GWAS and meta-analyses for ICP, examining >7 million common sequence variants in 1138 cases of ICP compared to 153,642 non-ICP controls. This resulted in the identification of 11 genome-wide significant loci associated with ICP. The application of a functional prioritization pipeline further enabled the identification of liver-enriched genes and liver-specific *cis*-regulatory elements as likely effectors of ICP risk.

## Results and discussion

**GWAS and GWAS meta-analyses for ICP.** GWAS were first performed using the NIHR-RD (https://bioresource.nihr.ac.uk/using-our-bioresource/our-cohorts/rare-diseases/) and 100KGP (https://www.genomicsengland.co.uk/initiatives/100000-genomes-project) data separately (Supplementary Fig. 1). In the NIHR-RD, 303 patients with ICP (severe, early-onset disease with symptoms by 33 weeks' gestation and maternal TSBA ≥ 40 µmol/L) were recruited. For women that used hormonal contraception, 15/254 (5.9%) reported itching when used and 39/256 reported cyclical itching. In addition, 15/247 patients (6.1%) reported drug-induced itching (other than for contraception). In the 100KGP, 225 individuals with an International Classification of Diseases (ICD) code for ICP were identified across all 100KGP study cohorts. For each of these datasets, common (minor allele frequency (MAF) ≥0.01) high-quality variants were extracted from the whole-genome sequence data. An unrelated subset of individuals of genetically defined European (EUR) ancestry (Supplementary Fig. 2), was identified by kinship coefficient estimation and principal component analysis. After quality control, the NIHR-RD dataset comprised 216 cases and 8436 controls with 8,291,828 variants, and the 100KGP dataset comprised 182 cases and 45,585 controls with 9,429,238 variants. A GWAS was performed in each cohort

separately using a generalized mixed model association test with saddle-point approximation to adjust for case–control imbalance using SAIGE[13] with sex and ten principal components (PCs) as covariates. The resulting summary statistics for variants with matching alleles and allele frequencies between the two datasets were then meta-analyzed using METAL[14], weighting the effect size estimates using the inverse of the standard errors. Variants showing heterogeneity of effect between the two datasets ($P < 1 \times 10^{-5}$) were excluded. In total, 8,199,999 variants were meta-analyzed and four loci were identified as achieving genome-wide significance ($P < 5 \times 10^{-8}$) (Supplementary Figs. 1, 3a). The genomic inflation (lambda) was 1.01 (Supplementary Fig. 3c), indicating no significant population stratification.

To enhance the power for genetic discovery, we meta-analyzed the NIHR-RD/100KGP (UK) ICP GWAS meta-analysis summary statistics with GWAS data from FinnGen (Release 4) (https://www.finngen.fi/en). FinnGen is a public-private partnership combining digital health record data from Finnish health registries with genotyping data from Finnish Biobanks. Release 4 includes association data at 16,962,023 variants for 2444 endpoints in 176,899 Finnish individuals. As with the 100KGP, cases of ICP were identified based on ICD codes. There were 740 cases and 99,621 controls following FinnGen phenotype evaluation. GWAS was performed by FinnGen using SAIGE[13] with sex, age, 10 PCs, and genotyping batch as covariates. Analysis of the summary statistics for the FinnGen ICP GWAS revealed seven loci achieving genome-wide significance (Supplementary Figs. 1, 3b, d), including the four loci identified in the UK meta-analysis. After allele matching, allele frequency matching, and removal of heterogeneous effects, 7,715,762 variants were available for meta-analysis across all three cohorts. In total, 11 loci achieved the genome-wide significance threshold ($P < 5 \times 10^{-8}$), (Fig. 1, Table 1, and Supplementary Fig. 1) and the genomic inflation (lambda) for the overall meta-analysis was 1.027 (Supplementary Fig. 3e). Summary statistics for the lead variants in each of the separate GWAS and meta-analyses are provided in Supplementary Data 1.

Conditional and joint (GCTA-COJO)[15] analyses in each of the genome-wide significant loci demonstrated a single signal underlying each association. Testing for epistatic interactions was performed across all 55 possible pairs of the 11 risk variants in NIHR-RD and 100KGP (for which full genotype data were available) separately with PLINK[16] and the resulting P values were combined using Fisher's method. No significant interactions were identified after correcting for multiple testing based on 55 comparisons ($P < 0.0009$). Post hoc power calculations based on the size of the cohort utilized for the combined meta-analysis demonstrated 80% power to detect effects with an odds ratio (OR) > 1.3 at MAF = 0.5 and OR > 1.7 at MAF ≥ 0.05 (Supplementary Fig. 4).

**Functional fine-mapping of causal variants at ICP susceptibility loci.** The liver is a central organ in the development of ICP, as evidenced by the fact that most known ICP causal mutations affect liver-specific genes[8,9] (Supplementary Fig. 5a). Moreover, unbiased analysis of the 11 ICP association signals from our GWAS highlighted that the expression of genes within these loci is highly liver-specific (Supplementary Fig. 5b) and relate to pathways involved in bile acid and lipid metabolism (Supplementary Fig. 6). We, therefore, reasoned that both coding and noncoding ICP risk variants likely affect liver-specific genes and/or *cis*-regulatory elements (CREs). In order to gain insights into the genetic mechanisms driving ICP susceptibility and identify likely causal variants, we carried out functional fine-mapping of the 11 association signals detected in the meta-analysis using PAINTOR[17] to calculate the probability of a variant being causal

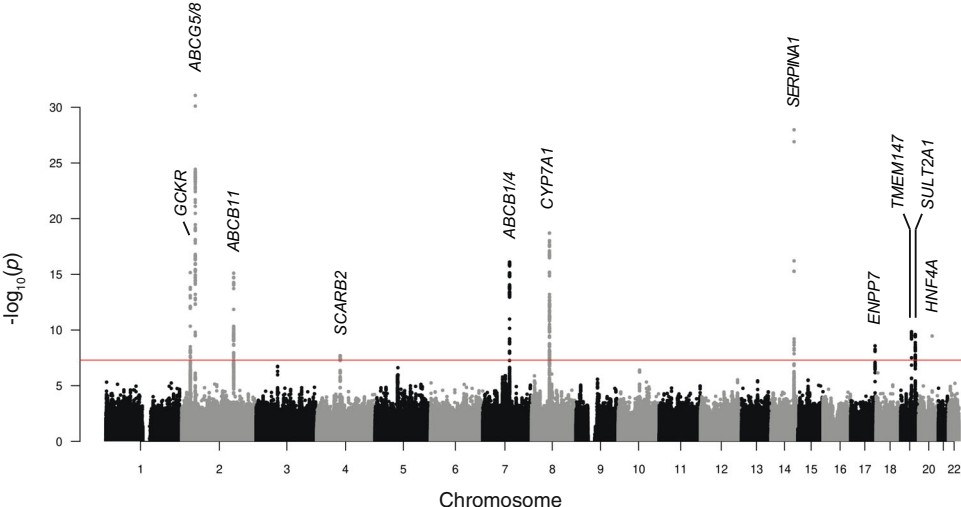

**Fig. 1 Manhattan plot for the genome-wide association study meta-analysis of intrahepatic cholestasis of pregnancy (ICP).** Data shown are from the combined meta-analysis including 1138 cases and 153,642 controls, all of European ancestry. The chromosomes are ordered on the x axis; the y axis shows the $-\log_{10}(P)$ values for the association tests. Eleven loci achieved genome-wide significance ($P < 5 \times 10^{-8}$, indicated by the red line). The prioritized gene at each locus is shown. The effector gene functional prioritization strategy followed in this study is presented in Fig. 2. Association testing was performed using a generalized logistic mixed model to account for population stratification and saddle-point approximation to control for type 1 error rates due to unbalanced case–control ratios, followed by meta-analysis weighting the effect size estimates using the inverse of the standard errors.

**Table 1 Association statistics of the eleven loci achieving genome-wide significance ($P < 5 \times 10^{-8}$) in the genome-wide association study meta-analysis of intrahepatic cholestasis of pregnancy (ICP).**

| Chr | Position | Variant (rsID) | Non-risk allele | Risk allele | UK Freq | Finn Freq | OR | 95% CI | P |
|---|---|---|---|---|---|---|---|---|---|
| 2 | 27508073 | rs1260326 | T | C | 0.598 | 0.650 | 1.43 | 1.35–1.52 | 6.84E-16 |
| 2 | 43844604 | rs4148211 | G | A | 0.608 | 0.556 | 1.70 | 1.61–1.78 | 1.72E-33 |
| 2 | 169074756 | rs66927685 | TGAACCTTAGAGACTGAAGAA | T | 0.455 | 0.430 | 1.42 | 1.33–1.5 | 7.83E-16 |
| 4 | 76490987 | rs13146355 | G | A | 0.451 | 0.456 | 1.27 | 1.19–1.36 | 2.00E-08 |
| 7 | 87500928 | rs5885586 | TG | T | 0.821 | 0.833 | 1.63 | 1.51–1.74 | 7.83E-17 |
| 8 | 58480178 | rs10107182 | T | C | 0.341 | 0.377 | 1.50 | 1.41–1.58 | 1.95E-19 |
| 14 | 94378610 | rs28929474 | C | T | 0.021 | 0.020 | 7.09 | 6.74–7.43 | 1.04E-28 |
| 17 | 79732281 | rs34491636 | A | G | 0.217 | 0.231 | 1.36 | 1.26–1.46 | 2.62E-09 |
| 19 | 35552195 | rs2251250 | C | T | 0.379 | 0.321 | 1.34 | 1.25–1.42 | 1.41E-10 |
| 19 | 47867143 | rs296384 | G | T | 0.837 | 0.847 | 1.46 | 1.34–1.58 | 2.50E-10 |
| 20 | 44413724 | rs1800961 | C | T | 0.030 | 0.045 | 2.06 | 1.84–2.29 | 3.42E-10 |

Chromosome (Chr), position (Build 38), and rsID are given for the lead variant (variant with the lowest P value) at each locus. For each variant, the risk allele has been indicated (the allele that is more common in cases than controls) with the corresponding odds ratios (OR). The frequencies of the risk alleles in control individuals in the UK (UK Freq) and Finnish (Finn Freq) datasets are shown. Association testing was performed using a generalized logistic mixed model to account for population stratification and saddle-point approximation to control for type 1 error rates due to unbalanced case–control ratios, followed by meta-analysis weighting the effect size estimates using the inverse of the standard errors.

at a given locus considering not only its strength of association but also functional annotation data (Fig. 2). We included adult liver tissue active chromatin features in the datasets used to prioritize causal variants, including accessible chromatin regions and those enriched in H3K27ac (see "Methods"). This analysis enabled the prioritization of nine variants with a high functional posterior probability ($PP_{func} > 0.8$) across eight ICP susceptibility loci (Supplementary Data 2). For the remaining three loci, which did not include any protein-altering variant, we assumed that variants residing within active liver CREs were more likely functional. We thus defined a high-confidence set of human liver accessible chromatin sites using ATAC-seq profiles from four whole liver samples (data from ENCODE[18], see "Methods"), which allowed us to further prioritize one likely causal variant at each remaining locus (Table 2).

**ICP coding variant analysis.** Three ICP signals (*SERPINA1*, *GCKR*, and *HNF4A*) hosted a single missense coding variant with

high likelihood of being causal, with functional posterior probabilities above 99% and a pathogenicity score CADD >10[19] (Fig. 3). From these three signals, the one with the strongest association with ICP susceptibility was rs28929474 in the *SERPINA1* gene, which encodes alpha-1-antitrypsin, a major plasma serine protease inhibitor (Fig. 3a). This variant has been extensively reported in the literature as the *SERPINA1* Z allele, an amino acid substitution of Glu342Lys that causes alpha-1-antitrypsin deficiency[20] and is associated with cystic fibrosis liver disease[21]. The two other missense variants alter the sequences of the *GCKR* (glucokinase regulatory protein) and *HNF4A* (hepatic nuclear factor 4 alpha) (Fig. 3b, c), genes that are essential for hepatic metabolic homeostasis and when mutated may lead to monogenic diabetes[22]. We examined the potential functional impact of these coding variants through the querying of the ClinVar variant interpretation database and Metadome analysis[23]. At *GCKR*, we identified an ICP risk variant, rs1260326 (Leu446Pro), which has been previously associated with altered

serum fasting plasma glucose and triglyceride concentrations[24,25] and nonalcoholic fatty liver disease (NAFLD)[26]. Previous functional studies of this variant have indicated that the risk allele in this locus is associated with increased glucokinase activity in the liver[27]. At *HNF4A*, a single variant was associated with ICP susceptibility, rs1800961 (Thr139Ile), which Metadome analysis[23] predicted to be deleterious for protein function (Supplementary Fig. 7a).

**Functional prioritization of noncoding variants at lipid and bile acid homeostasis loci.** At the remaining loci identified in the GWAS meta-analysis for ICP, we observed that seven of the eleven loci contained noncoding and synonymous variants (examining the 95% genetic credible set, as well as the lead variant and those in linkage equilibrium at $r^2 > 0.8$ in EUR individuals). This observation is in line with previous reports suggesting that common sequence variation predominantly contributes to human disease via disruption of noncoding *cis*-regulatory elements[28,29]. We therefore took advantage of the functional fine-mapping strategy described above, along with the integration of adult liver epigenomic datasets and pathogenicity measures, to further dissect the causal mechanisms at play in each ICP locus (Fig. 2, see "Methods").

In the liver, the family of ABC transporters play central roles in numerous physiological processes, including the export of cholesterol, bile salts, bilirubin, and drug conjugates. Three ABC transporter gene loci (*ABCG5/8*, *ABCB1/4*, and *ABCB11*) contained ICP GWAS signals (Fig. 4). *ABCG5* and *ABCG8* together encode a heterodimeric cholesterol transporter, expressed in hepatocytes and intestinal cells[30]. A missense polymorphism in *ABCG8*, Tyr54Cys (rs4148211), was identified as the lead variant at this locus (Fig. 4a). However, Metadome analysis[23] indicated that this protein sequence change is expected to be tolerated (Supplementary Fig. 7b and Supplementary Data 3), suggesting that this or other variants in linkage disequilibrium (LD) have a different type of functional impact at this locus. Combined analysis of adult liver epigenomic datasets and a high-throughput reporter assay dataset from liver cells (Survey of Regulatory Elements, SuRE[31]), indicated that the lead variant rs4148211 ($PP_{func} > 0.99$) resides within a hepatic CRE and its risk allele is associated with allele-dependant transcriptional activity in liver cells ($P = 9.1 \times 10^{-4}$), where the risk allele is associated with transcriptional repression (Fig. 4b and Supplementary Fig. 7c). Analysis of the SuRE assay dataset from liver cells revealed seven additional ICP-associated variants in the *ABCG8* locus for which the risk allele associates with

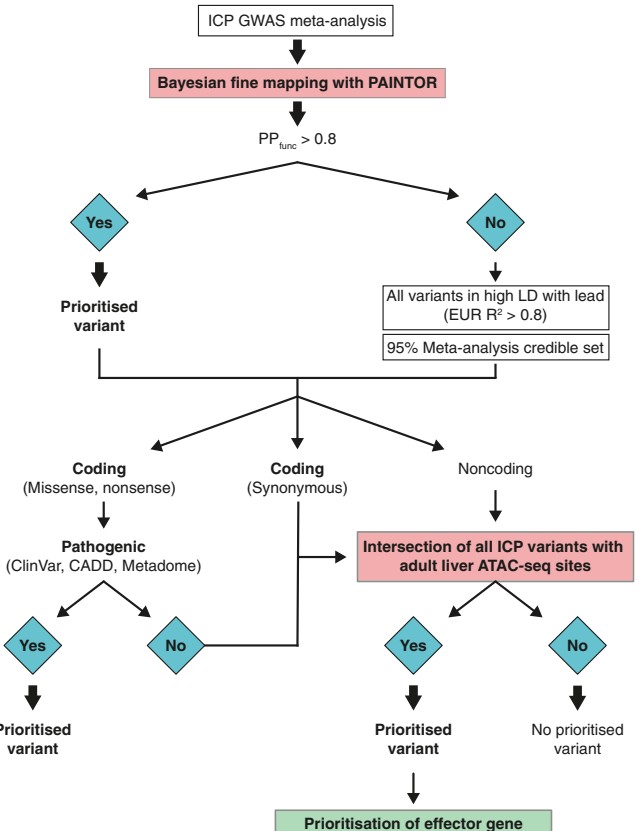

**Fig. 2 Flowchart describing the variant prioritization strategy followed for the eleven loci reaching genome-wide significance in the GWAS meta-analysis of ICP.** Sequential steps used to prioritize the variants are shown with the different approaches for nonsynonymous, synonymous, and noncoding variants indicated.

**Table 2 Prioritized variants and genes within each of the eleven ICP genome-wide significant loci.**

| | Lead variant | | | Functional prioritization | | | | |
|---|---|---|---|---|---|---|---|---|
| Chr | Position | rsID | Gene | Variant (rsID) | Same as lead | Non-risk | Risk | Functional impact |
| 2 | 27508073 | rs1260326 | *GCKR* | rs1260326 | Yes | T | C | Coding, missense |
| 2 | 43844604 | rs4148211 | *ABCG5/8* | rs4148211 | Yes | G | A | CRE |
| 2 | 169074756 | rs66927685 | *ABCB11* | rs66927685 | Yes | TGAACCTTAGAGACTGAAGAA | T | CRE |
| 4 | 76490987 | rs13146355 | *SCARB2* | rs4859682 | No | C | A | CRE |
| 7 | 87500928 | rs5885586 | *ABCB1/4* | rs55747905 | No | C | T | CRE, creation of TCF7L2-binding site |
| 8 | 58480178 | rs10107182 | *CYP7A1* | rs10504255 | No | A | G | CRE, disruption of DMRTA1-binding site, and decreased gene expression |
| 14 | 94378610 | rs28929474 | *SERPINA1* | rs28929474 | Yes | C | T | Coding, missense |
| 17 | 79732281 | rs34491636 | *ENPP7* | rs9916601 | No | C | T | CRE, increased gene expression |
| 19 | 35552195 | rs2251250 | *TMEM147* | rs4806173 | No | C | G | CRE, creation of NC3R1-binding site and decreased gene expression |
| 19 | 47867143 | rs296384 | *SULT2A1* | rs296361 | No | G | A | CRE, creation of SOX-D-binding site and decreased gene expression |
| 20 | 44413724 | rs1800961 | *HNF4A* | rs1800961 | Yes | C | T | Coding, missense |

The chromosome (Chr), position (Build 38), and rsID are given for the lead variant (variant with the lowest *P* value) in each locus are shown along with the gene, rsID, risk, and non-risk alleles and functional impact following in silico functional prioritization.

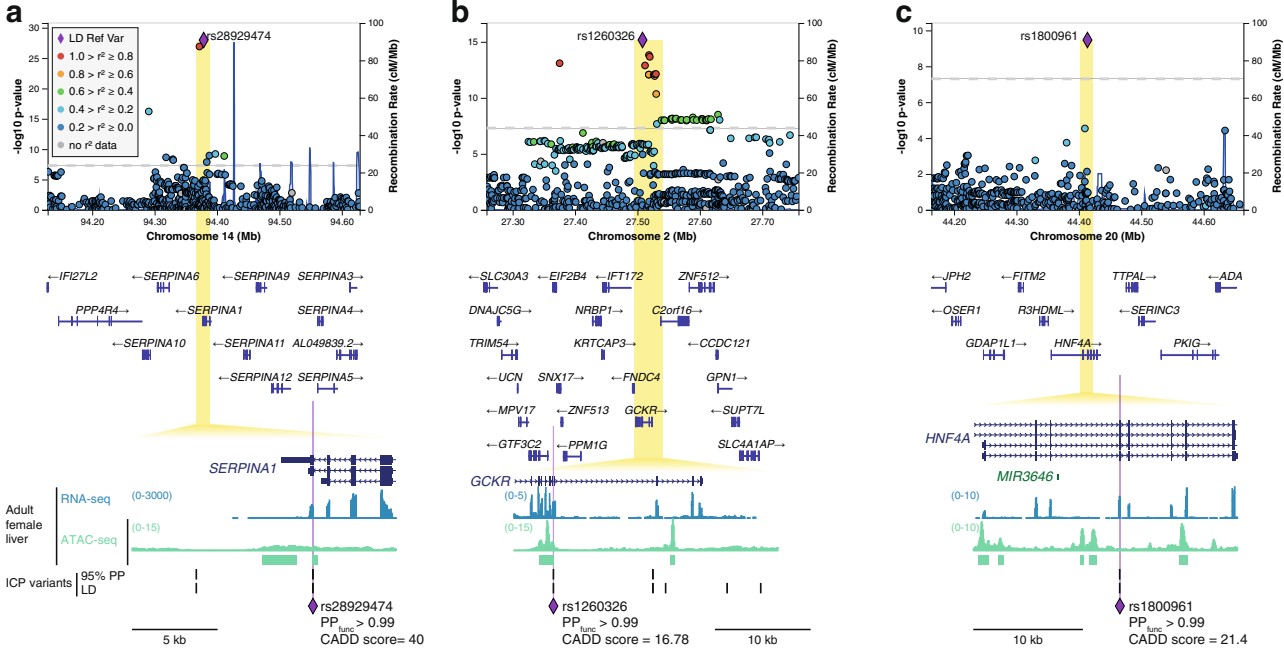

**Fig. 3 Analysis of intrahepatic cholestasis of pregnancy (ICP) association signals uncovers three coding risk variants affecting the *SERPINA1, GCKR*, and *HNF4A* genes. a–c** Regional association and linkage disequilibrium (LD) plots for the ICP signals at *SERPINA1, GCKR*, and *HNF4A*, respectively. The y axis represents the $-\log_{10}(P)$ values for the association test from the GWAS meta-analysis of ICP (1138 cases and 153,642 controls) and the x axis the chromosomal positions (GRCh38/hg38). The SNP with the lowest p-value in the locus is indicated by a purple diamond. The remaining SNPs in the region are colored according to their $r^2$ with the lead SNP ($r^2$ calculated using 1000 Genomes European (EUR) reference). The dotted gray line represents the genome-wide significance threshold of $5 \times 10^{-8}$. Plots were generated using LocusZoom[61]. Association testing was performed as described in Fig. 1. Zoomed insets reveal the prioritized coding variant at each locus, with functional posterior probability ($PP_{func}$ above 0.99. $PP_{func}$ were calculated using PAINTOR[17]. All three variants showed Combined Annotation Dependent Depletion (CADD) scores[19,84] above 10, which is usually considered the threshold for pathogenic variants. Further details on coding variant analysis are provided in Supplementary Data 3. 95% PP ICP variants, variants within the 95% genetic credible set in the ICP meta-analysis. LD ICP variants, variants with EUR $r^2 > 0.8$ with the meta-analysis lead variant.

transcriptional repression (Supplementary Fig. 7c). However, only one overlapped a hepatic CRE (rs4148204) and none of them was prioritized as likely causal by PAINTOR (Supplementary Data 2).

The ICP association signal on chromosome 7 spanned a genomic segment including the genes *ABCB1* and *ABCB4* (Fig. 4c), which encode for the Multidrug Resistance Proteins 1 and 3 (MDR1 and MDR3), respectively. Whilst the lead variant in this locus did not overlap a coding sequence or active hepatic CRE, we prioritized the variant rs55747905 as likely functional ($PP_{func} = 0.94$), which overlapped an active enhancer element (accessible chromatin enriched in H3K27ac) (Fig. 4c). Transcription factor binding motif disruption analysis further revealed that the risk allele of rs55747905 (C) disrupts a recognition sequence for the Wnt signaling transcription factor TCF7L2 (Fig. 4d).

At the third ABC transporter locus, we prioritized *ABCB11* as the gene mediating the ICP susceptibility process. In this case, functional fine-mapping with PAINTOR enabled the prioritization of a single variant residing in an intronic liver enhancer in the *DHRS9* gene (Fig. 4e). However, analysis of ENCODE[18] RNA-seq data revealed that *DHRS9* has very weak expression in adult liver, in contrast with *ABCB11*, which encodes the bile salt export pump (BSEP).

The ICP association signal on chromosome 8 is located between the *UBXN2A* and *CYP7A1* genes (Fig. 5a). *UBXN2A* encodes a ubiquitin-like protein involved in proteasomal degradation[32] and *CYP7A1* encodes the enzyme (cholesterol 7 alpha-hydroxylase) responsible for the first and rate-limiting step in bile acid biosynthesis in hepatocytes[33]. At this locus, only one regulatory variant was prioritized, rs10504255, which resides in

an active liver enhancer. Unlike the lead variant at this locus (rs10107182), the prioritized variant reached a high functional posterior probability ($PP_{func} > 0.99$ vs. <0.01) and had a high pathogenicity score (CADD = 12.9 vs. 5.8). Furthermore, analysis of GTEx[34] eQTL data revealed that rs10504255 is associated with allele-dependant expression of *UBXN2B* in human liver (Fig. 5b), where the risk allele (G) is associated with weaker transcriptional activity ($P = 6.18 \times 10^{-6}$). Even though this variant was not a statistically significant eQTL for *CYP7A1* in the GTEx database ($P = 0.15$), we observed a similar trend on the direction of effect of the risk allele (Fig. 5b) and previous chromatin conformation capture and genome editing studies in human hepatocytes have linked this CRE with the modulation of *CYP7A1* expression[35]. These previous studies, combined with the functional significance of CYP7A1 in bile acid homeostasis, led us to prioritize *CYP7A1* as the major effector transcript in this locus. Motif analysis further revealed that the risk allele disrupts a binding site for the transcription factor DMRTA1 (Fig. 5c).

In addition to *CYP7A1*, the ICP GWAS meta-analysis identified another important locus for bile acid homeostasis, *SULT2A1*, which encodes the enzyme Sulfotransferase 2A1, highly expressed in the liver and capable of phase II metabolism of bile acids[36] (Fig. 5d). Here, a variant in the promoter of *SULT2A1* was prioritized (rs296361), with the risk allele conferring lower *SULT2A1* expression in the liver ($P = 1.31 \times 10^{-7}$) (Fig. 5e), likely through the creation of a binding site for the family of transcriptional repressors SOX-D (Fig. 5f). We note that the prioritized variant also associated with allele-dependent effects on the expression of a nearby long noncoding RNA (*LINC01595*, $P = 1.02 \times 10^{-50}$), albeit in the

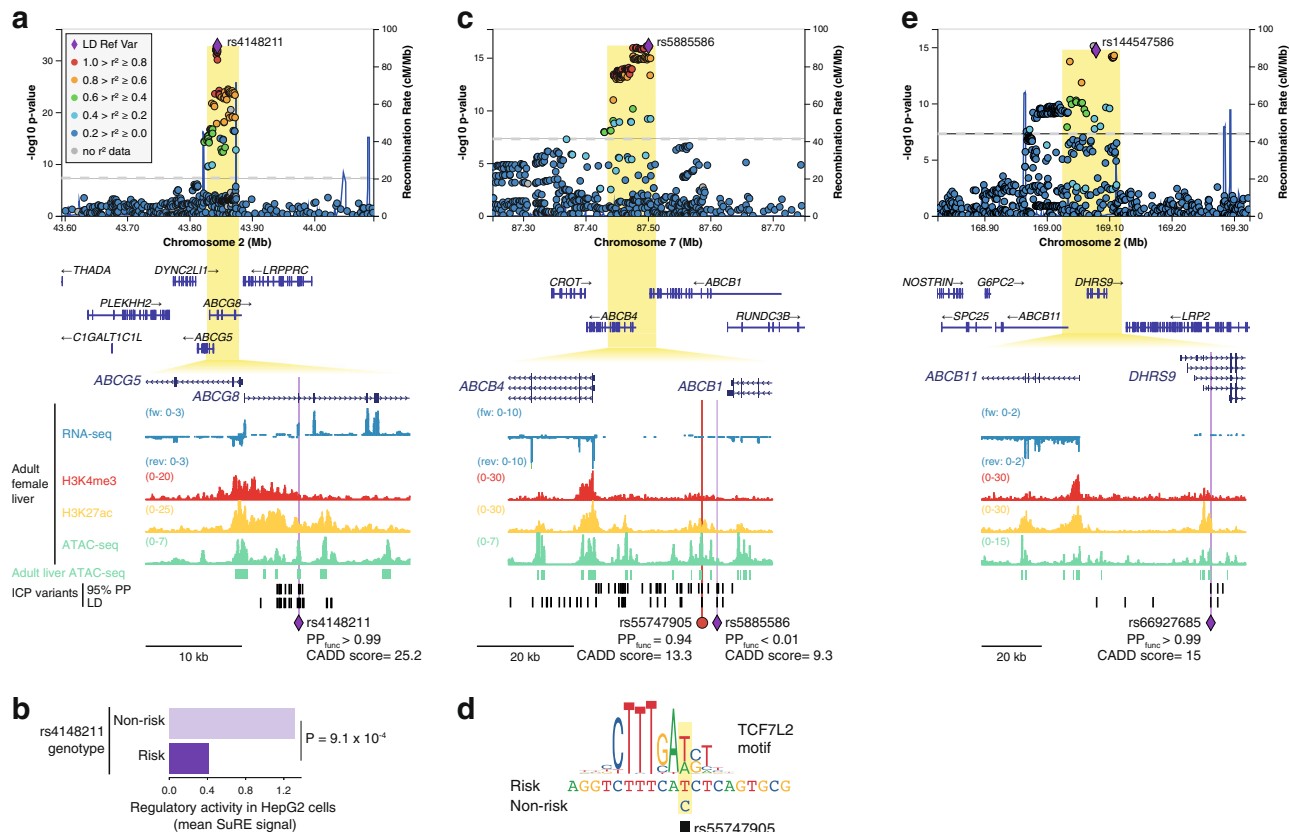

**Fig. 4 Three ABC transporter gene loci are associated with intrahepatic cholestasis of pregnancy (ICP). a, c, e** LocusZoom[61] plots showing the association signals at *ABCG8*, *ABCB1/4*, and *DHRS9*, respectively. The LocusZoom plots are as per Fig. 3 and association testing was performed as described in Fig. 1. At each locus, a single noncoding variant was prioritized as likely causal following the strategy outlined in Fig. 2. All epigenomic and transcriptomic datasets shown were retrieved from ENCODE[18]. Adult liver ATAC-seq peaks track (green regions) corresponds to regions accessible in at least two out of four ENCODE adult liver ATAC-seq (see Methods). 95% PP ICP variants, variants within the 95% genetic credible set in the ICP meta-analysis. LD ICP variants, variants with EUR $r^2 > 0.8$ with the meta-analysis lead variant. PP$_{func}$, functional posterior probability calculated using PAINTOR[17]. CADD, combined annotation-dependent depletion. **a** Zoomed inset shows a coding variant at *ABCG8* overlapping a hepatic transcriptional enhancer, as revealed by strong H3K27ac enrichment and accessible chromatin in adult female liver tissue. Metadome analysis did not suggest that this variant affects protein function (see Supplementary Fig. 7). **b** Allele-specific effect of the prioritized risk variant rs4148211, assessed by Survey of Regulatory Elements (SuRE) in HepG2 cells[31]. The results suggest that the risk variant associates with decreased transcriptional activity. Mean signal and two-sided Wilcoxon rank-sum test. Previously determined 5% false-discovery rate threshold: $P < 0.00173121$[31]. Source data are provided as a Source Data file. **c, e** Zoomed insets show two prioritized regulatory risk variants for ICP: one affecting a hepatic enhancer between *ABCB1* and *ABCB4* (**c**), and one affecting a hepatic enhancer in an intron of *DHRS9* (**d**). We note however that *DHRS9* is not expressed in liver tissue, as shown in the strand-specific RNA-seq tracks. In contrast, *ABCB11* is highly expressed in this tissue making it a likely effector transcript in this locus.

opposite direction. This discrepancy may reflect an indirect effect of the expression of *SULT2A1* on *LINC01595*.

**Functional prioritization of previously uncharacterized ICP genes.** We next explored the remaining three ICP-associated signals where the closest genes were not directly involved in lipid and bile acid homeostasis. On chromosome 19, even though it was not possible to employ PAINTOR for functional fine-mapping due to the presence of ambiguous alleles (A/T or G/C), we prioritized the variant rs4806173 based on its overlap with a hepatic CRE (Fig. 6a). This variant resides in an intron of the testis-specific gene *GAPDHS*[37]. We thus queried eQTL GTEx[34] data to identify potential targets of this regulatory variant in liver tissue. This analysis led us to pinpoint *TMEM147* (transmembrane protein 147) as the likely effector transcript of this association signal, with the risk allele associating with lower hepatic gene expression ($1.52 \times 10^{-40}$) (Fig. 6b). Using transcription factor motif disruption analysis, we detected that the risk allele of rs4806173 creates a recognition sequence for the glucocorticoid

receptor (Fig. 6c), which has been previously reported to act as either a gene expression activator or repressor in a context-dependent manner[38,39]. Recent studies in HeLa cells have linked loss of *TMEM147* with increased cholesterol uptake[40] supporting the role of this gene as an effector of ICP genetic susceptibility.

Similarly, we prioritized *ENPP7* as the likely effector transcript driving the association signal in chromosome 17 (Fig. 6d, e), with the risk variant (T) associating with increased *ENPP7* expression (Fig. 5e). *ENPP7* encodes the alkaline sphingomyelinase (Alk-SMase) that is present in the intestinal tract and bile, and responsible for the digestion of sphingomyelins in a bile salt-dependent manner[41]. Previous studies have suggested that Alk-SMase promotes intestinal cholesterol absorption[42].

The final signal encompassed intronic variants in the *SHROOM3* gene, including one variant in a hepatic CRE (rs4859682) (Fig. 7a). However, analysis of liver eQTLs did not assist in the identification of potential effector transcripts in the locus. Topologically associating domains (TADs) correspond to self-interacting genomic domains that contain genes and regulatory elements that interact more frequently with each

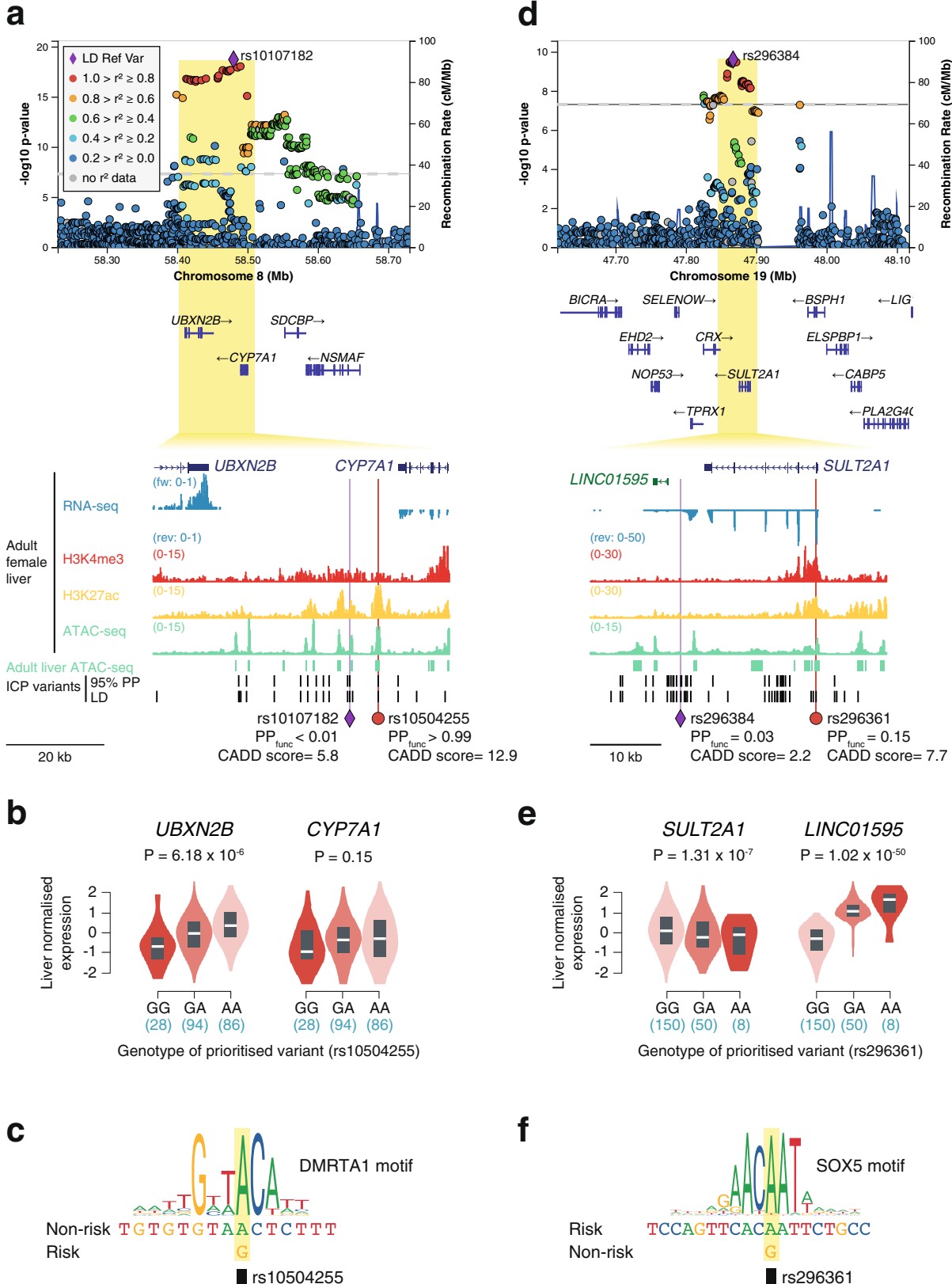

other than with other regions[43]. We thus carried out a systematic analysis of all genes located in the TAD that contains this ICP association signal, interrogating their expression in liver tissue (Fig. 7b) and their liver-specificity (Fig. 7c). Given the liver-specificity of most ICP genes (Supplementary Fig. 5b), we decided to further investigate genes whose expression is both high and

specific in liver tissue. Two genes met these criteria: *STBD1* and *SCARB2* (Fig. 7c). We then investigated human liver Hi-C maps in order to find out whether there were long-range 3D chromatin contacts between the prioritized causal variant rs4859682, *SHROOM3* (closest gene), *STBD1,* and *SCARB2*, in this tissue. Only long-range interactions stemming from *SCARB2* were

**Fig. 5 GWAS meta-analysis of ICP reveals association signals affecting the bile acid homeostasis genes *CYP7A1* and *SULT2A1*. a, d** LocusZoom[61] plots showing the association signals at *CYP7A1* and *SULT2A1*, respectively. The LocusZoom plots are as per Fig. 3 and association testing was performed as described in Fig. 1. All epigenomic and transcriptomic datasets shown were retrieved from ENCODE[18]. Adult liver ATAC-seq peaks track (green regions) corresponds to regions accessible in at least two out of four ENCODE adult liver ATAC-seq (see "Methods"). 95% PP ICP variants, variants within the 95% genetic credible set in the ICP meta-analysis. LD ICP variants, variants with EUR $r^2 > 0.8$ with the meta-analysis lead variant. $PP_{func}$, functional posterior probability calculated using PAINTOR. CADD, combined annotation-dependent depletion. **a** Functional fine-mapping with PAINTOR[17] identified a single variant (rs10504255) with a posterior probability over 0.99 overlapping an active liver enhancer located between the *CYP7A1* gene, which encodes the rate-limiting enzyme for bile acid synthesis, and *UBXN2B*. **b, e** GTEx[34] [GTEx Analysis Release V8 (dbGaP Accession phs000424.v8.p2)] human liver eQTLs identify the top prioritized variant in the *CYP7A1* locus (rs10504255) as an eQTL associated with the eGene *UBXN2B* (**b**) and show that the *SULT2A1* promoter ICP variant (rs296361) is an eQTL for both *SULT2A1* and its downstream lncRNA *LINC01595* (**e**). Violin plots represent the density distribution of the samples in each genotype (n for each genotype is indicated below in blue). Box plots show normalized gene expression in median (white line), first and third quartiles. Gene-level adjusted P values calculated with FastQTL[85] are shown. **c** The risk allele of the ICP-associated variant rs10504255 (G) disrupts a DMRTA1-binding motif. We note that this allele also associates with weaker transactivation in liver tissue, as observed by eQTL analysis (**b**). **d** Overlap of ICP risk variants with adult liver accessible chromatin sites revealed one variant in the promoter of *SULT2A1* (rs296361). **f** Transcription factor motif analysis revealed that the ICP variant rs296361 affects a motif for the SOX-D transcriptional repressor family.

---

detected (Fig. 7d), which led us to prioritize this gene as the likely effector of the association signal in this locus. *SCARB2* encodes for the lysosomal integral membrane protein type 2 (LIMP-2), which has been reported to play a role in cholesterol transport in lysosomes[44].

**Overlap of risk loci for ICP and other traits**. All but one of the loci identified in this study (rs34491636, *ENPP7*) have been previously associated (at $r^2 > 0.6$) with diseases or traits in the GWAS Catalog[45], many of which have phenotypic overlap with ICP (Supplementary Fig. 8 and Supplementary Data 4). Notably, six of the loci have previously been associated with gallstone disease, six with LDL cholesterol concentrations, and four with liver enzyme concentrations (Supplementary Fig. 8). Likewise, we observed a strong enrichment for ICP susceptibility genes (identified in an unbiased manner from the GWAS loci) being also associated with gallstone disease ($P = 2.5 \times 10^{-15}$) and LDL cholesterol ($P = 2.67 \times 10^{-07}$) (Supplementary Fig. 9).

The elevated serum bile acids that form the core phenotype are also associated with maternal and fetal dyslipidemia[46–48], and increased rates of gallstone formation[49]. Variation at *ABCG8* has previously been linked to gallstone disease and other lipid-related phenotypes[50,51]. However, the specific haplotype associated with ICP differs from the main reported association with gallstone disease. *CYP7A1* variation is associated with blood lipid traits[52] in addition to gallstone formation[53]. Heterozygous *ABCB4* mutations were the first genetic link identified to ICP[54,55] and common variation at this locus has been previously described in ICP[10]. In the context of pregnancy, raised gestational hormones cause reduced FXR-mediated induction of hepatic Bsep, Shp, and Mdr3[6], and enterocyte Shp and Fgf15/19[56] thereby causing abnormal bile acid homeostasis. This is likely to exacerbate susceptibility to hypercholanemia in genetically predisposed women with the variants reported in this manuscript.

Given the strong overlap between genes associated with ICP and cholelithiasis (Supplementary Fig. 9), we directly examined the evidence of association of the 11 ICP loci with cholelithiasis using the FinnGen cholelithiasis GWAS (https://r4.finngen.fi/pheno/K11_CHOLELITH), which includes 15,683 cases and 158,425 controls. Six of the loci identified in our ICP GWAS meta-analysis achieve genome-wide significance for cholelithiasis as well ($P < 5 \times 10^{-08}$) (Supplementary Fig. 8). Two additional loci demonstrate nominal evidence of association. Despite 100% power (assuming a similar-sized effect on the risk of ICP and cholelithiasis) to detect association, three ICP loci (*ABCB11*, *SCARB2*, and *ENPP7*) showed no evidence of association with cholelithiasis indicating that they are specific to ICP and not shared with cholelithiasis.

We also directly examined evidence of association at the 29 loci with AF > 0.01 (the MAF threshold in our study) previously associated with cholelithiasis in the largest meta-analysis published to date (comprising 27,174 cholelithiasis cases and 736,838 controls[57]). Five of these loci achieve genome-wide significance ($P < 5 \times 10^{-08}$) in the ICP meta-analysis (*SERPINA1*, *CYP7A1*, *GCKR*, *ABCB4*, and *HNF4A*) and a further five show evidence of association with ICP after correcting for examining 29 loci ($P < 0.05/29 = 0.0017$); two of these are genes reported as genome-wide significant in the present ICP GWAS (*ABCG8* and *SULT2A1*) and three are not (*FADS2*, *JMJD1C*, and *TTC39B*). Assuming the same effect size of these associations between cholelithiasis and ICP, the power to detect association at $P = 0.0017$ was only present for one locus (at which significance was achieved) so we examined whether a composite effect of the known cholelithiasis genetic risk variants was detectable in ICP patients by calculating polygenic risk scores (PRS). Of the 29 cholelithiasis loci, genotype data were available in the NIHR-RD dataset for 27. PRS calculated using the OR from the previously published cholelithiasis GWAS demonstrated ICP patients have a significantly higher burden of cholelithiasis genetic risk variants compared with controls (Wilcoxon $P = 1.4 \times 10^{-05}$). However, when this analysis was repeated with the above loci that showed association in the ICP meta-analysis removed (leaving 18 variants for which data were available), there was no difference in PRS ($P = 0.198$) between ICP and controls, despite 92% power to detect a difference at $P < 0.05$. This suggests that not all the genetic risk factors for cholelithiasis are risk factors for ICP. Together, these findings indicate that cholelithiasis and ICP have distinct but overlapping sets of genetic risk factors.

Given the overlapping genetic architecture between ICP and gallstone disease, and the broad spectrum of disease severity in this group, these loci represent possible modifier genes in a number of other cholestatic diseases such as the progressive familial cholestasis syndromes[58].

In conclusion, this study has uncovered common variation that influences ICP susceptibility, identifying key contributing genes. These genes highlight an underlying shared mechanism of variation at liver-specific genes and *cis*-regulatory elements, providing insights into the pathogenesis of this disease that warrant further investigation.

## Methods

**Study cohorts and ethics**. An extensive description of the cohorts included in this study and details of the ethics approvals can be found in the Supplementary Note in Methods (Supplementary Information).

**NIHR-RD and 100KGP whole-genome sequencing variant calling, quality control, ancestry, and relatedness estimation**. Detailed methods describing the processing of whole-genome sequence data for the NIHR-RD and 100KGP cohorts

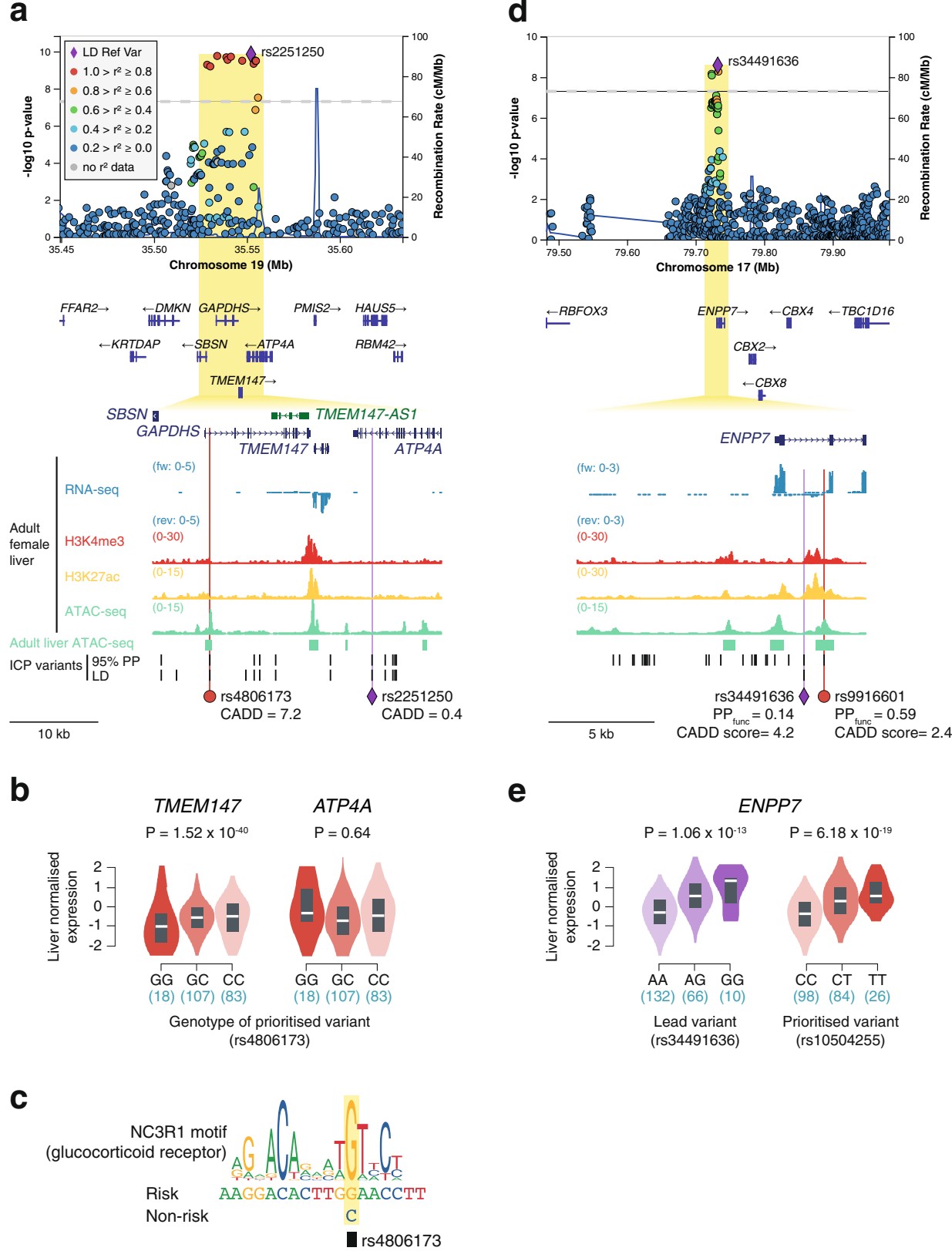

can be found in the Supplementary Note in Methods (Supplementary Information).

**NIHR-RD and 100KGP genome-wide association analyses.** For genome-wide association analysis (GWAS) in both the NIHR-RD and 100KGP cohorts

separately, variants passing the following criteria were retained: MAF ≥ 0.01, minimum minor allele count (MAC) ≥ 20, missingness <1%, Hardy–Weinberg equilibrium $P$ value $>1 \times 10^{-6}$ and differential (case/control) missingness $P$ value $>1 \times 10^{-5}$. The final NIHR-RD and 100KGP datasets comprised 8,337,027 and 9,545,879 variants, respectively. GWAS was performed in each dataset separately using the R package SAIGE[13] (versions 0.42.1 and 0.44.2) (https://github.com/

**Fig. 6 Analysis human liver eQTLs uncovers *TMEM147* and *ENPP7* as effector transcripts of ICP susceptibility. a, d** LocusZoom[86] plots showing the association signals at *TMEM147* and *ENPP7*, respectively. The LocusZoom plots are as per Fig. 3 and association testing was performed as described in Fig. 1. All epigenomic and transcriptomic datasets shown were retrieved from ENCODE[18]. Adult liver ATAC-seq peaks track (green regions) corresponds to regions accessible in at least two out of four ENCODE adult liver ATAC-seq (see "Methods"). 95% PP ICP variants, variants within the 95% genetic credible set in the ICP meta-analysis. LD ICP variants, variants with EUR $r^2 > 0.8$ with the meta-analysis lead variant. PP_func, functional posterior probability calculated using PAINTOR[17]. CADD, combined annotation-dependent depletion. Overlap of ICP-associated variants in these two loci with adult liver accessible chromatin sites enabled the prioritization of two ICP regulatory variants in intronic regions of *GAPDHS* (rs4806173) (**a**) and *ENPP7* (rs9916601) (**d**). **b, e** The ICP variants prioritized with functional liver chromatin annotations are eQTLs in liver tissue and pinpoint *TMEM147* and *ENPP7* as effector transcripts in these loci. Violin plots represent the density distribution of the samples in each genotype (n for each genotype is indicated below in blue). Box plots show normalized gene expression in median (white line), first and third quartiles. Gene-level adjusted *P* values calculated with FastQTL[86] are shown. Liver eQTL plots were retrieved from GTEx[34] [GTEx Analysis Release V8 (dbGaP Accession phs000424.v8.p2)]. We note that *GAPDHS* is a testis-specific gene and its interrogation as an eGene for rs4806173 was therefore not possible (**b**). **c** The ICP risk allele of rs4806173 (G), which associates with less transcription of *TMEM147*, is predicted to create a binding site for glucocorticoid receptors, a family of context-dependent transcriptional regulators that can act as either activators or repressors[87].

weizhouUMICH/SAIGE). SAIGE implements a generalized logistic mixed model to account for population stratification and a saddle-point approximation to control type 1 error rates when case–control ratios are unbalanced. SAIGE was run using the default parameters. Null logistic mixed models were fitted using the subsets of linkage disequilibrium (LD)-free high-quality common SNVs. Sex and the top ten principal components were used as fixed effects. The genomic inflation factors (lambda), calculated based on the 50th percentile, were 1.009 and 1.001 in NIHR-RD and 100KGP, respectively, indicating no evidence of confounding by population structure. The summary statistics from the NIHR-RD were lifted over from build GRCh37 to build GRCh38 of the human genome using CrossMap[59] (version 0.5.2) (http://crossmap.sourceforge.net/). In all, 12,756 variants failed to lift over. Due to data sharing restrictions, it was not possible to combine the raw genome-wide genotype data from the NIHR-RD and 100KGP datasets.

**NIHR-RD and 100KGP meta-analysis.** Meta-analysis of the NIHR-RD and 100KGP GWAS summary statistics was performed using METAL (version 2011-03-25) (https://genome.sph.umich.edu/wiki/METAL)[14]. Multi-allelic variants were excluded such that there were 8,291,828 and 9,545,879 variants in NIHR-RD and 100KGP, respectively. Of these, 8,207,819 variants were shared and 8,201,630 variants had matching alleles. Meta-analysis was performed weighting the effect size estimates using the inverse of the standard errors. Variants showing heterogeneity of effect between the two datasets ($P < 1 \times 10^{-5}$) and those in which the minimum/maximum allele frequencies differed by >0.05 were excluded leaving 8,199,999 variants. A Manhattan plot was produced using the R package qqman (version 0.1.8) (https://github.com/stephenturner/qqman)[60] (Supplementary Figure 3a). Four loci achieved genome-wide significance ($P < 5 \times 10^{-8}$). A QQ plot was generated using the observed and expected –log10 (*P* values) (Supplementary Fig. 3c). The genomic inflation factor (lambda), calculated based on the 50th percentile, was 1.01 indicating no significant population stratification.

**FinnGen genome-wide association analysis.** FinnGen is a public-private partnership combining digital health record data from Finnish health registries with genotyping data from Finnish Biobanks. Release 4 (https://finngen.gitbook.io/documentation/v/r4/) includes association data at 16,962,023 variants for 2444 endpoints in 176,899 Finnish individuals. A detailed description of the study design and analytical methods are available in the online documentation (URL above). In brief, individuals were genotyped with Illumina and Affymetrix chip arrays. QC was performed to remove samples and variants of poor quality. Genome-wide imputation was performed using reference Finnish whole-genome sequence data. Disease endpoints were defined using nationwide registries based on ICD (and other) codes. A subset of unrelated individuals of genetically confirmed Finnish ancestry was identified. GWAS was performed using SAIGE (version 0.35.8.8) for all variants with MAC > 5. Sex, age, 10 PCs, and genotyping batch were included as covariates in the analysis.

For the ICP GWAS (https://r4.finngen.fi/pheno/O15_ICP), cases ($n = 740$) were identified as those with the ICD-10 code of O26.6 ("Liver disorders in pregnancy, childbirth and the puerperium" including "cholestasis (intrahepatic) in pregnancy" and "obstetric cholestasis") or an ICD-9 code of 646.7 ("Liver and biliary tract disorders in pregnancy"). All females without these ICD codes were included as controls ($n = 99,621$). Seven loci achieved genome-wide significance ($P < 5 \times 10^{-8}$) (Supplementary Fig. 3b). The genomic inflation factor (lambda), calculated based on the 50th percentile, was 1.046 (Supplementary Fig. 3d).

**Combined UK (NIHR-RD and 100KGP) and FinnGen meta-analysis.** A meta-analysis was subsequently performed of the combined NIHR-RD and 100KGP data ("UK meta-analysis") and the FinnGen summary statistics. This included 154,780 individuals of whom 1138 were cases and 153,642 were controls. Meta-analysis was performed using the same methodology as described above. A total of 7,721,597 variants were shared between the final UK meta-analysis dataset and the FinnGen

dataset and 7,715,997 variants had matching alleles. After performing the meta-analysis, variants showing heterogeneity of effect between the two datasets ($P < 1 \times 10^{-5}$) and those in which the minimum/maximum allele frequencies differed by >0.25 were excluded leaving 7,715,762 variants. Eleven loci achieved genome-wide significance ($P < 5 \times 10^{-8}$) (Fig. 1). The genomic inflation factor (lambda), calculated based on the 50th percentile, was 1.027 (Supplementary Fig. 3e). Summary statistics for each of the lead variants in each of the separate GWAS and meta-analyses are provided in Supplementary Data 1.

**LocusZoom.** Regional high-resolution association plots showing the LD between markers in the prioritized loci (as per European reference data) were generated using LocusZoom (version 0.13.3) (https://my.locuszoom.org/)[61]. Credible sets were calculated by LocusZoom using Bayes factors based on the *P* values (https://statgen.github.io/gwas-credible-sets/method/locuszoom-credible-sets.pdf).

**Epistasis.** Testing for epistatic interactions was performed across all possible 55 pairs of the 11 prioritized lead risk variants in NIHR-RD and 100KGP separately with PLINK[16] (version 1.9) (https://www.cog-genomics.org/plink/). The resulting *P* values were combined using Fisher's method. No significant interactions were identified after correcting for multiple testing based on 55 comparisons ($P < 0.0009$).

**Conditional analysis.** Each of the prioritized loci was examined using "conditional and joint analyses" (GCTA-COJO)[15] to assess whether there was more than one independent signal. COJO was run from GCTA (version 1.93.2) (https://cnsgenomics.com/software/gcta/#COJO) using default parameters (examining a 10-Mb window surrounding genome-wide significant variants). The NIHR-RD dataset was utilized to provide the reference LD structure of the variants.

**Power calculations.** Post hoc power calculations were performed to assess the range of odds ratio (OR) and allele frequencies for which associations could be detected at genome-wide significance ($P < 5 \times 10^{-8}$) given the number of cases and controls in the final meta-analysis. The R package genpwr[62] (version 1.0.4) (https://cran.r-project.org/web/packages/genpwr/index.html) was employed using a logistic model under genetic additivity. Power to detect specific previously described associations was calculated with genpwr using the OR of the described associations, reference European AF and cohort sizes. For cholelithiasis, the OR were taken from the supplementary data of a published meta-analysis[57] (Supplementary Data 1). For rs708686 the OR from the additive model described in the manuscript was utilized.

**Polygenic risk score calculation.** A polygenic risk score (PRS) for cholelithiasis in the NIHR-RD data was calculated using Mangrove (version 1.21) (https://cran.r-project.org/web/packages/Mangrove/index.html)[63] with the previously published cholelithiasis OR[57] and reference European AF, as described above. The PRS was calculated assuming additivity within and between loci. Statistical comparison of the PRS in cases and controls was performed using an unpaired two-sample two-sided Wilcoxon rank-sum test.

Power calculations for the PRS analyses were performed through simulating genotype data for case and control datasets using custom code, written in R (available on request). Genotypes were simulated for each variant in each individual with the probabilities of each genotype status (homozygous reference, heterozygous, and homozygous alternate) determined by the Hardy–Weinberg equilibrium formulae based on the AF and assuming complete linkage equilibrium between the variants. For controls, the reference European AF were used. For cases, the AF of the risk allele were calculated from the reference AF and OR. A total of 100 repetitions of the simulation process were performed in which genotypes for the target variants were generated for the required number of cases and controls

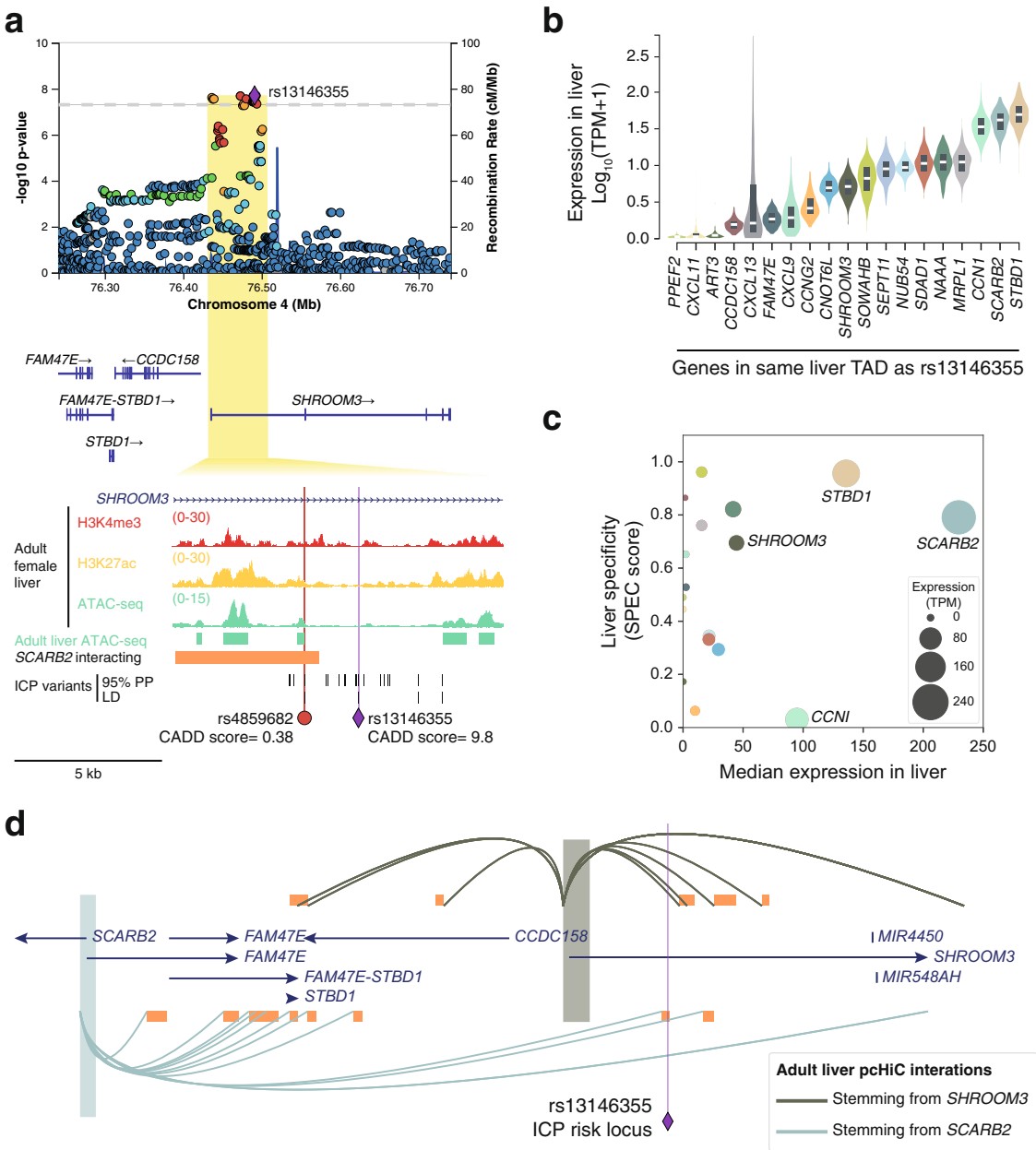

**Fig. 7 Tissue-specificity and 3D chromatin conformation analyses prioritize SCARB2 as an ICP susceptibility gene. a** LocusZoom[86] plot showing the association signal at *SHROOM3*. The LocusZoom plots are as per Fig. 3 and association testing was performed as described in Fig. 1. All epigenomic and transcriptomic datasets shown were retrieved from ENCODE[18]. Adult liver ATAC-seq peaks track (green regions) corresponds to regions accessible in at least two out of four ENCODE adult liver ATAC-seq (see "Methods"). 95% PP ICP variants, variants within the 95% genetic credible set in the ICP meta-analysis. LD ICP variants, variants with EUR $r^2 > 0.8$ with the meta-analysis lead variant. CADD, combined annotation-dependent depletion. Overlap of ICP risk variants in this locus with adult liver accessible chromatin sites enabled the prioritization of a single ICP-regulatory variant in an intronic region of the *SHROOM3* gene (rs4859682). None of the other ICP variants (EUR $r^2 > 0.8$ with lead variant or in the 95% PP credible set) overlapped accessible chromatin sites. **b** Expression of all genes within the TAD containing the ICP lead variant rs13146355 in the human liver. Expression data was retrieved from GTEx[34] [GTEx Analysis Release V8 (dbGaP Accession phs000424.v8.p2)]. Violin plots represent the density distribution of expression of the indicated genes in human liver tissue ($n = 226$ donors). Box plots show gene expression in median (white line), first and third quartiles. TPM, transcripts per million. Note that the gene harboring the ICP risk variant, *SHROOM3*, is lowly expressed in the liver. **c** Scatter plot showing the expression of all genes within the TAD containing the ICP lead variant rs13146355 in the human liver versus their liver-specificity. Tissue-specificity scores (SPECS) were precalculated by Everaert et al.[81]. Source data are provided as a Source Data file. This plot highlights two highly expressed and liver-specific genes, *STBD1*, and *SCARB2*. **d**, Analysis of long-range chromatin interactions in human liver detected by promoter capture Hi-C[79], showing that the genomic region containing the ICP risk variant rs4859682 interacts with *SCARB2*, but not *SHROOM3*. Long-range chromatin interactions stemming from the promoter of *STBD1* were not detected in liver tissue.

(matching the numbers available in the ICP analysis). For each repetition, PRS were generated for each individual using Mangrove, and the statistical significance of the difference between cases and controls was calculated using Wilcoxon rank-sum test. The power to detect a difference was then determined as the percentage of simulations in which a statistically significant difference between cases and controls was seen at $P < 0.05$.

**Unbiased gene prioritization and expression/pathway/GWAS Catalog analyses**. An unbiased assessment of gene expression and functional pathway enrichment of the genes within the 11 prioritized risk loci was undertaken as follows. All 655 variants achieving $P < 5 \times 10^{-8}$ in the final meta-analysis were identified. Of these, 629 were successfully mapped to an rsID. All variants in LD with these variants at $r^2 > 0.8$ in European individuals using 1000 Genomes data were identified using LDlinkR (version 1.1.2) (https://cran.r-project.org/web/packages/LDlinkR/vignettes/LDlinkR.html)[64]. This resulted in the identification of 871 unique variants. These variants were subsequently analyzed using the Ensembl Variant Effect Predictor (VEP)[65] (version 104) (https://www.ensembl.org/info/docs/tools/vep/index.html) to identify the 39 unique genes impacted. These genes were interrogated using the GENE2FUNC tool in FUMA (version 1.3.6b) (https://fuma.ctglab.nl/gene2func)[66]. 35 genes were successfully mapped and analyzed by GENE2FUNC (*ABCB1, ABCB11, ABCB4, ABCG5, ABCG8, ATP4A, BABAM2, C2orf16, CCDC121, CRX, CYP7A1, DHRS9, EIF2B4, ENPP7, GAPDHS, GCKR, GPN1, HNF4A, LINC01595, MRPL33, NSMAF, PPIAP85, PPP4R4, RBKS, SBSN, SDCBP, SERPINA1, SERPINA10, SERPINA2, SERPINA6, SHROOM3, SNX17, SULT2A1, TMEM147, TMEM147-AS1, TPRX2P, UBXN2B, ZNF512, and ZNF513*). The results of tissue-specificity analysis (as per GTEx[34] data (https://www.gtexportal.org)), gene sets/pathway analysis (as per MsigDB[67] (https://www.gsea-msigdb.org/gsea/msigdb)) and overlap of the genes with GWAS Catalog[45] reported associations (https://www.ebi.ac.uk/gwas/) were interrogated. Further details regarding the methodology employed by GENE2FUNC are available on the FUMA website (https://fuma.ctglab.nl/tutorial#g2fOutputs). Statistical significance was assessed using hypergeometric tests with Bonferroni correction for multiple testing.

**Overlap with previous GWAS findings**. Previously described GWAS findings (as per the GWAS Catalog (https://www.ebi.ac.uk/gwas)[45] accessed on July 23, 2021) were searched for overlap with the lead variants at each of the 11 genome-wide significant ICP loci. This was undertaken using the LDtrait Tool from LDlink (https://ldlink.nci.nih.gov/?tab=ldtrait)[68]. All associations achieving $P < 5 \times 10^{-6}$ that were in LD with the 11 identified lead variants at $r^2 > 0.6$ in European populations (using 1000 Genomes data) were included. Phenotype labels were manually amended such that they were consistent and duplicates were removed (Supplementary Fig. 8).

**Identification of adult human liver replicated accessible regions**. In order to functionally annotate ICP risk variants, we retrieved from the ENCODE portal[18] (https://www.encodeproject.org/) the IDR ranked peaks for a set of high-quality ATAC-seq experiments carried out on four adult human liver samples. The file identifiers were ENCFF948WBQ, ENCFF012SCX, ENCFF658LHI, and ENCFF658LHI, and corresponded to liver samples from two males and two females with ages ranging from 40 to 61 years old (mean 47 years old). Using these files, we identified the set of genomic regions that were accessible in at least two of the four donors, which we used to prioritize and identify functional ICP regulatory variants. These regions are provided in Supplementary Data 5 (coordinates from genome assembly GRCh38/hg38).

**Variant functional prioritization**. The overall prioritization strategy to identify the most likely risk causal variants and genes at each of the ICP-associated loci is presented in Fig. 2. Details of each of the steps taken to prioritize causal variants and genes are described in the next sections.

*Statistical fine-mapping.* We applied PAINTOR (version 3.1) (https://bogdan.dgsom.ucla.edu/pages/paintor/)[17] (a Bayesian fine-mapping approach) which uses an empirical Bayes prior to integrate functional annotation data, LD patterns and strength of association to estimate the posterior probability (PP) of a variant being causal. Variants within a 100-kb window centering on the lead variant at each of the 11 genome-wide significant loci in the meta-analysis were included. LD matrices of pairwise correlation coefficients were derived using European 1000 Genomes (Phase 3) imputed data[69], excluding variants with ambiguous alleles (A/T or G/C) where reference/alternate alleles could not be reliably matched. Each variant was intersected with the following functional annotations to generate a PP of being causal (PP_func): PhastCons elements (phastConsElements100way, updated August 5, 2015), ENCODE[18] ATAC-seq replicated peaks generated from the adult liver (see above), and Roadmap E066 (adult liver) ChIP-seq narrow peaks for H3K27ac and H3K4me1 histone modifications. Variants with PP_func > 0.8 were prioritized for further investigation.

For the loci in which PAINTOR did not identify at least one variant with a high posterior probability of being causal (PP_func > 0.8), we then proceeded to intersect the coordinates of all ICP variants in the locus with human adult liver accessible chromatin sites (see details above) using pybedtools version 0.8.0[70] (https://daler.

github.io/pybedtools/). This approach was taken because none of the remaining loci contained coding variants. To carry out this analysis, we first defined ICP variants as either (a) all variants within the 95% credible set defined in the ICP GWAS meta-analysis by LocusZoom, or (b) all variants in high LD ($r^2 > 0.8$) with the meta-analysis lead variant in European (EUR) individuals. High LD variants were identified using LDlink (https://ldlink.nci.nih.gov/?tab=ldproxy).

*Variant functional annotation of coding variants.* ICP risk variants were first annotated in relation to their potential to disrupt protein-coding sequences. We identified four ICP-associated loci in which there was at least one coding non-synonymous risk variant. To investigate their potential pathogenicity, we first interrogated them in ClinVar (https://www.ncbi.nlm.nih.gov/clinvar/)[71]. For the ICP coding variants not reported as pathogenic in ClinVar, we performed further pathogenicity analysis using MetaDome[23] (version 1.0.1), which is based on the concept of protein domain homology in the human genome. Briefly, homologous Pfam protein domains were aggregated into meta-domains. Then, population variation from the Exome Aggregation Consortium (ExAC) (https://exac.broadinstitute.org/) and pathogenic mutations from the Human Gene Mutation Database (HGMD)[72] (http://www.hgmd.cf.ac.uk/ac/index.php) were used to create genetic tolerance profiles across human meta-domains at amino acid resolution. MetaDome genetic tolerance profiles were previously derived using 56,319 transcripts, 71,419 protein domains, 12,164,292 genetic variants from gnomAD[73], and 34,076 pathogenic mutations from ClinVar[71] and queried via the online portal at https://stuart.radboudumc.nl/metadome. Interrogation of the coding variant in the *GCKR* gene was not possible due to a lack of data availability. Finally, we retrieved the CADD scores of pathogenicity (release version 1.6, GRCh38/hg38) for all coding variants from https://cadd.gs.washington.edu/download. The pathogenicity analysis results are presented in Supplementary Data 3.

*Analysis of the survey of regulatory elements data.* Survey of Regulatory Elements was previously performed in HepG2 cells[31]. In that study, the authors interrogated 5.9 million human biallelic SNPs for their ability to confer allele-dependent regulatory activity. We retrieved this dataset (SuRE_SNP_table_LP190708.txt.gz) from https://osf.io/pjxm4/ and queried all ICP variants (see details above) at each of the 11 ICP risk loci. Data are presented as non-risk versus risk allele. $P$ values were obtained by a two-sided Wilcoxon rank-sum test, comparing for each SNP the set of SuRE values of all fragments containing the reference allele versus the set of SuRE values of all fragments containing the alternative allele. Following the original report[31], $P < 0.00173121$ were considered as passing a 5% false-discovery rate threshold. Data mining was performed using Python (version 3.7) and figures were plotted using seaborn[74] version 0.11.2 (https://seaborn.pydata.org/).

*Transcription factor binding motif analysis.* The prioritized ICP-associated variants (see details above) were investigated for their potential effect on TF binding affinity using Motifbreak R version 2.2.0 (https://bioconductor.org/packages/release/bioc/html/motifbreakR.html). To query motifs, we imported MotifDb (https://bioconductor.org/packages/release/bioc/html/MotifDb.html) (version 1.30.0), which contains >4200 TF motifs, including >2800 motifs from studies in human samples and includes the databases HOCOMOCO[75], HOMER[76], and ENCODE-motifs[18]. For visual representation, motif logos were downloaded from JASPAR 2020[77] (http://jaspar.genereg.net/).

*Analysis of human liver eQTLs.* Human liver eQTL data shown in this study were retrieved from the Genotype-Tissue Expression (GTEx) Portal[34] on May 12, 2021, which at the time of accession contained data for 208 liver samples. [GTEx Analysis Release version 8 (dbGaP Accession phs000424.v8.p2)]. At each ICP association signal, we queried the lead variant and, if different, the variant that was functionally prioritized. The analysis was restricted to liver eQTLs and eGenes contained in the same TAD as the lead variant (human liver TAD coordinates were retrieved from http://3dgenome.fsm.northwestern.edu/publications.html). Figures were directly downloaded from the GTEx Portal.

*Analysis of adult liver 3D chromatin conformation.* Gene expression and eQTL analyses were restricted to the topologically associating domains (TADs) containing the ICP-associated variants identified in our meta-analysis. Human liver TAD coordinates (GRCh38/hg38)[78] were retrieved from http://3dgenome.fsm.northwestern.edu/downloads/hg38.TADs.zip.

For specific investigation of long-range chromatin interactions in liver tissue, we interrogated two complementary datasets from adult liver: Hi-C[78] and capture Hi-C[79]. These datasets were visualized and analyzed using the 3D-genome interaction viewer and database (http://www.3div.kr/).

**Visualization of human adult liver datasets**. Human adult female liver (right lobe) epigenomic and transcriptomic datasets we downloaded from the ENCODE portal[18] (https://www.encodeproject.org/) in bigwig format and visualized using the UCSC Genome Browser[80] (GRCh38/hg38). All data corresponds to fold change over control for the shown assay. The visualized files had the following identifiers: ENCFF232QBB (ATAC-seq), ENCFF280QYJ (H3K4me3 ChIP-seq) and ENCFF853WPX (H3K27ac ChIP-seq). Adult liver 3D chromatin interaction

datasets were visualized and analyzed using the 3D-genome interaction viewer and database (http://www.3div.kr/).

**Tissue-specificity analysis.** In order to investigate whether the ICP risk variants at *SHROOM3* could be in the vicinity of or interact with liver-specific genes, we used SPECS scores[81], which were pre-computed for adult liver for all Ensembl (GRCh38.v85) genes using all GTEx[34] samples. In brief, SPECS is a non-parametric tissue-specificity score that is compatible with unequal sample group sizes. SPECS uses all individual data points available and enables the detection of features that are specifically present or absent in one or more tissue types. The SPECS score has been shown to outperform other tissue-specificity scores[81], including z-score[82] and JSD[83]. Data mining was performed using Python (v3.7) and figures were plotted using seaborn[74] version 0.11.2 (https://seaborn.pydata.org/)[18].

**Reporting summary.** Further information on research design is available in the Nature Research Reporting Summary linked to this article.

## Data availability

The genome-wide summary statistics for the final combined meta-analysis carried out in this study have been deposited in the GWAS Catalog under accession code GCP000309, available at https://www.ebi.ac.uk/gwas/studies/GCST90095084. The lead variants' GWAS and GWAS meta-analysis summary statistics generated in this study are provided in Supplementary Data 1. Genotype and phenotype data from the NIHR-RD participants are available from several sources. 4,835 of the NIHR-RD participants were part of the 100,000 Genomes Project—Rare Diseases Pilot. These data can be accessed by application to Genomics England Limited following the procedure outlined at https://www.genomicsengland.co.uk/about-gecip/joining-researchcommunity/. The genotype and phenotype data from the remaining 7348 NIHR-RD participants can be accessed by application to the NIHR BioResource Data Access Committee at dac@bioresource.nihr.ac.uk. Subject to ethical consent, the genotype data of a subset of 6,939 NIHR-RD participants are also available from the European Genome-phenome Archive (EGA) at the EMBL European Bioinformatics Institute [https://ega-archive.org/dacs/EGAC00001000259]. This includes data from 305 ICP cases [https://ega-archive.org/datasets/EGAD00001004515]. Genomic and phenotype data from the 100KGP participants can be accessed by application to Genomics England Limited following the procedure outlined at https://www.genomicsengland.co.uk/about-gecip/joining-researchcommunity/. Genotype data for the 764 UK Biobank samples are available through the UK Biobank [https://www.ukbiobank.ac.uk/]. The FinnGenn GWAS summary statistics are publicly accessible following registration [https://www.finngen.fi/en/access_results]. The adult human liver replicated accessible regions generated in this study are provided in Supplementary Data 5. The variant pathogenicity data used in this study is available in the ClinVar database [https://www.ncbi.nlm.nih.gov/clinvar/] and through MetaDome v1.0.1 [https://stuart.radboudumc.nl/metadome]. The CADD scores (v1.6) used in this study are available on the CADD website [https://cadd.gs.washington.edu/download]. The liver epigenomic and transcriptomic data used in this study are available in the ENCODE portal[18] [https://www.encodeproject.org/]. The SuRE data[31] used in this study are available in the OFS data repository [https://osf.io/pjxm4/]. The transcription factor motif logos used in this study are available in the JASPAR database [http://jaspar.genereg.net/]. The eQTL data used in this study are available in the Genotype-Tissue Expression (GTEx) Portal[34] under dbGaP accession phs000424.v8.p2 [https://gtexportal.org/home/]. The tissue-specificity SPECS[81] scores for liver tissue used in this study are available in the SPECS web browser [https://specs.cmgg.be]. The adult liver TAD coordinates used in this study are available in the 3D Genome Browser [http://3dgenome.fsm.northwestern.edu/index.html].

## Code availability

Custom code used in this study is available upon request.

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

## Acknowledgements

We thank NIHR BioResource volunteers for their participation, and gratefully acknowledge NIHR BioResource Centres, NHS Trusts and staff for their contribution. We thank the National Institute for Health Research, NHS Blood and Transplant, and Health Data Research UK as part of the Digital Innovation Hub Programme. P.H.D. and C.W. are supported by the National Institute for Health Research (NIHR) Biomedical Research Centre at Guy's and St Thomas' Foundation Trust and King's College London. A.P.L. is supported by an NIHR Academic Clinical Lectureship. C.W. is funded by an NIHR Senior Investigator award. The views expressed are those of the authors and not necessarily those of the NHS, NIHR, or Department of Health and Social Care. I.C. is an Academy of Medical Sciences Springboard Fellow (SBF005\1050). The Scheme is supported by the British Heart Foundation, Diabetes UK, the Global Challenges Research

Fund, the Government Department for Business, Energy and Industrial Strategy and the Wellcome Trust. I.C. was also supported by the Foundation of National Institutes of Health (FNIH) and the Accelerated Medicines Partnerships Type 2 Diabetes (AMP T2D) initiative (RFP16) and by the National Institute for Health Research (NIHR) Biomedical Research Centre at Imperial College Healthcare NHS Trust. We also thank the Imperial College London High-Performance Computing Service. MMYC is supported by a Kidney Research UK Clinical Training Fellowship. D.P.G. is supported by the St Peter's Trust. The Genotype-Tissue Expression (GTEx) Project was supported by the Common Fund of the Office of the Director of the National Institutes of Health, and by NCI, NHGRI, NHLBI, NIDA, NIMH, and NINDS. The data used for the analyses described in this manuscript were obtained from the GTEx Portal on May 12, 2021. This research was also made possible through access to the data and findings generated by the 100,000 Genomes Project. The 100,000 Genomes Project is managed by Genomics England Limited (a wholly owned company of the Department of Health and Social Care). The 100,000 Genomes Project is funded by the National Institute for Health Research and NHS England. The Wellcome Trust, Cancer Research UK, and the Medical Research Council have also funded research infrastructure. The 100,000 Genomes Project uses data provided by patients and collected by the National Health Service as part of their care and support. The authors gratefully acknowledge the participation of the patients and their families recruited to the 100,000 Genomes Project. We want to acknowledge the participants and investigators of the FinnGen study.

## Author contributions

Study conception and design: P.H.D., A.P.L., I.C., M.C., D.P.G., and C.W. Laboratory work: A.S.A., A.A., A.L.M., and J.Z. Bioinformatics: P.H.D., A.P.L., I.C., M.C., M.M., D.P.G., and H.M. Patient recruitment and phenotyping: J.C., A.S., J.D., S.C., M.G., K.N., M.T., W.M.H., M.C.E., and H.U.M. All authors critically reviewed the manuscript and approved the final version.

## Competing interests

The authors declare no competing interests.

## Additional information

## NIHR BioResource

Julian Adlard[16], Munaza Ahmed[17], Tim Aitman[18,19], Hana Alachkar[20], David Allsup[21], Jeff Almeida-King[22], Philip Ancliff[23], Richard Antrobus[24], Ruth Armstrong[25,26,27], Gavin Arno[28,29], Sofie Ashford[6,30], William Astle[6,30,31], Anthony Attwood[6,30], Chris Babbs[32,33], Tamam Bakchoul[34], Tadbir Bariana[35,36], Julian Barwell[37,38], David Bennett[39], David Bentley[40], Agnieszka Bierzynska[41], Tina Biss[42], Marta Bleda[43], Harm Bogaard[44], Christian Bourne[40], Sara Boyce[45], John Bradley[6], Gerome Breen[46,47], Paul Brennan[48,49], Carole Brewer[50], Matthew Brown[6,30], Michael Browning[51], Rachel Buchan[52,53], Matthew Buckland[54], Teofila Bueser[55,56,57], Siobhan Burns[54], Oliver Burren[43], Paul Calleja[58], Gerald Carr-White[56], Keren Carss[6,30], Ruth Casey[25,26,27], Mark Caulfield[7], John Chambers[59,60], Jennifer Chambers[8,9], Floria Cheng[9], Patrick F. Chinnery[6,61,62], Martin Christian[63], Colin Church[64], Naomi Clements Brod[6,30], Gerry Coghlan[54], Elizabeth Colby[41], Trevor Cole[65], Janine Collins[66], Peter Collins[67], Camilla Colombo[40], Robin Condliffe[68], Stuart Cook[52,69,70,71], Terry Cook[72], Nichola Cooper[73], Paul Corris[74,75], Abigail Crisp-Hihn[6,30], Nicola Curry[76], Cesare Danesino[77], Matthew Daniels[78,79], Louise Daugherty[6,30], John Davis[6,30], Sri V. V. Deevi[6,30], Timothy Dent[79], Eleanor Dewhurst[6,30], Peter Dixon[1], Kate Downes[6,30], Anna Drazyk[80], Elizabeth Drewe[81], Tina Dutt[82], David Edgar[83], Karen Edwards[6,30], William Egner[84], Wendy Erber[85], Marie Erwood[6,30], Maria C. Estiu[14], Gillian Evans[86], Dafydd Gareth Evans[87], Tamara Everington[88], Mélanie Eyries[89], Remi Favier[90,91,92], Debra Fletcher[6,30], James Fox[6,30], Amy Frary[6,30], Courtney French[93], Kathleen Freson[94], Mattia Frontini[6,30], Daniel Gale[2], Henning Gall[95], Claire Geoghegan[40], Terry Gerighty[40], Stefano Ghio[96], Hossein-Ardeschir Ghofrani[73,95], Simon Gibbs[52], Kimberley Gilmour[97], Barbara Girerd[98,99,100], Sarah Goddard[101], Keith Gomez[35,36], Pavels Gordins[102], David Gosal[20], Stefan Gräf[6,30,43], Luigi Grassi[6,30], Daniel Greene[6,30,31], Lynn Greenhalgh[103], Andreas Greinacher[104], Paolo Gresele[105], Philip Griffiths[106,107], Sofia Grigoriadou[108], Russell Grocock[40], Detelina Grozeva[25], Scott Hackett[109], Charaka Hadinnapola[43],

William Hague[13], Matthias Haimel[6,30,43], Matthew Hall[81], Helen Hanson[103], Kirsty Harkness[110], Andrew Harper[52,78,111], Claire Harris[75], Daniel Hart[66], Ahamad Hassan[112], Grant Hayman[113], Alex Henderson[114], Jonathan Hoffmann[65], Rita Horvath[115,116], Arjan Houweling[44], Luke Howard[52], Fengyuan Hu[6,30], Gavin Hudson[115], Joseph Hughes[40], Aarnoud Huissoon[109], Marc Humbert[98,99,100], Sean Humphray[40], Sarah Hunter[40], Matthew Hurles[117], Louise Izatt[118], Roger James[6,30], Sally Johnson[119], Stephen Jolles[120,121], Jennifer Jolley[6,30], Neringa Jurkute[28,36], Mary Kasanicki[122], Hanadi Kazkaz[123], Rashid Kazmi[45], Peter Kelleher[54], David Kiely[68], Nathalie Kingston[6], Robert Klima[58], Myrto Kostadima[6,30], Gabor Kovacs[124,125], Ania Koziell[126,127], Roman Kreuzhuber[6,30], Taco Kuijpers[128,129], Ajith Kumar[17], Dinakantha Kumararatne[122], Manju Kuria[130,131], Michael Laffa[132,133], Fiona Lalloo[87], Michele Lamber[134,135], Hana Lango Alle[6,30], Allan Lawrie[136], Mark Layton[132], Claire Lentaigne[132,133], Adam Levine[2,3], Rachel Linger[6,30], Hilary Longhurst[108], Eleni Louka[32,33], Robert MacKenzie Ross[137], Bella Madan[138], Eamonn Maher[25,139], Jesmeen Maimaris[97], Sarah Mangles[140], Rutendo Mapeta[6,30], Kevin Marchbank[75], Stephen Marks[23], Hugh S. Markus[80], Hanns-Ulrich Marschall ![ORCID][15], Andrew Marshall[141,142,143], Jennifer Martin[6,30,43], Mary Mathias[144], Emma Matthews[36,145], Heather Maxwell[146], Paul McAlinden[75], Mark McCarthy[33,111,147], Stuart Meacham[6,34], Adam Mead[148], Karyn Megy[6,30], Sarju Mehta[149], Michel Michaelides[28], Carolyn Millar[132,133], Shahin Moledina[23], David Montani[98,99,100], Tony Moor[28,29], Nicholas Morrell[6,133], Monika Mozere[2], Keith Muir[150], Andrew Mumford[151,152], Michael Newnham[43], Jennifer O'Sullivan[138], Samya Obaji[67], Steven Okoli[32,33], Andrea Olschewski[124], Horst Olschewski[124,125], Kai Ren Ong[65], Elizabeth Ormondroy[78,79], Willem Ouwehan[6,30], Sofia Papadi[6,30], Soo-Mi Park[26,27,153], David Parry[19], Joan Paterson[25,26,27], Andrew Peacock[64], John Peden[40], Kathelijne Peerlinck[94], Christopher Penkett[6,30], Joanna Pepke-Zaba[154], Romina Petersen[6,30], Angela Pyle[115], Stuart Rankin[58], Anupama Rao[23], F. Lucy Raymond[6,25], Paula Rayner-Matthew[6,30], Christine Rees[40], Augusto Rendon[7], Tara Renton[58], Andrew Rice[155,156], Sylvia Richardson[31], Alex Richter[24], Irene Roberts[32,33,157], Catherine Roughley[86], Noemi Roy[32,33,157], Omid Sadeghi-Alavijeh[158], Moin Saleem[41], Nilesh Samani[159], Alba Sanchis-Juan[6,30], Ravishankar Sargur[84], Simon Satchell[41], Sinisa Savic[160], Laura Scelsi[96], Sol Schulman[161], Marie Scully[123], Claire Searle[162], Werner Seeger[95], Carrock Sewell[163], Denis Seyres[6,30], Susie Shapiro[76], Olga Sharmardina[6,30], Rakefet Shtoyerman[164], Keith Sibson[144], Lucy Side[17], Ilenia Simeoni[6,30], Michael Simpson[165], Suthesh Sivapalaratnam[66], Anne-Bine Skytte[166], Katherine Smith[7], Kenneth G. C. Smith[43,167], Katie Snape[168], Florent Soubrier[89], Simon Staines[6,30], Emily Staples[43], Hannah Stark[6,30], Jonathan Stephens[6,30], Kathleen Stirrups[6,30], Sophie Stock[6,30], Jay Suntharalingam[137], Emilia Swietlik[43], R. Campbell Tait[169], Kate Talks[42], Rhea Tan[80], James Thaventhiran[43], Andreas Themistocleous[39], Moira Thomas[170], Kate Thomson[78,79], Adrian Thrasher[23], Chantal Thys[94], Marc Tischkowitz[171], Catherine Titterton[6,30], Cheng-Hock Toh[82], Mark Toshner[43], Matthew Traylor[80], Carmen Treacy[43,154], Richard Trembath[55], Salih Tuna[6,30], Wojciech Turek[58], Ernest Turro[6,30,31], Tom Vale[38], Chris Van Geet[94], Natalie Van Zuydam[39], Marta Vazquez-Lopez[9], Julie von Ziegenweidt[6,30], Anton Vonk Noordegraaf[44], Quintin Waisfisz[44], Suellen Walker[23], James Ware[52,53,69], Hugh Watkins[78,79,111], Christopher Watt[6,30], Andrew Webster[28,29], Wei Wei[61], Steven Welch[109], Julie Wessels[101], Sarah Westbury[151,152], John-Paul Westwood[123], John Wharton[73], Deborah Whitehorn[6,30], James Whitworth[25,26,27], Martin R. Wilkins[73], Catherine Williamson ![ORCID][1,176]✉, Edwin Wong[107], Nicholas Wood[172,173], Yvette Wood[6,30], Geoff Woods[25,122], Emma Woodward[87], Stephen Wort[53,55], Austen Worth[23], Katherine Yates[6,30,43], Patrick Yong[174], Tim Young[6,30], Ping Yu[6,30] & Patrick Yu-Wai-Man[61]

[16]Chapel Allerton Hospital, Leeds Teaching Hospitals NHS Trust, Leeds, UK. [17]North East Thames Regional Genetics Service, Great Ormond Street Hospital for Children NHS Foundation Trust, London, UK. [18]MRC Clinical Sciences Centre, Faculty of Medicine, Imperial College London, London, UK. [19]Institute of Genetics and Molecular Medicine, University of Edinburgh, Edinburgh, UK. [20]Salford Royal NHS Foundation Trust, Salford, UK. [21]Queens Centre for Haematology and Oncology, Castle Hill Hospital, Hull and East Yorkshire NHS Trust, Cottingham, UK. [22]European

Molecular Biology Laboratory, European Bioinformatics Institute (EMBL-EBI), Wellcome Genome Campus, Hinxton, Cambridge, UK. [23]Great Ormond Street Hospital for Children NHS Foundation Trust, London, UK. [24]University Hospitals Birmingham NHS Foundation Trust, Birmingham, UK. [25]Department of Medical Genetics, Cambridge Institute for Medical Research, University of Cambridge, Cambridge Biomedical Campus, Cambridge, UK. [26]Cancer Research UK Cambridge Centre, Cambridge Biomedical Campus, Cambridge, UK. [27]NIHR Cambridge Biomedical Research Centre, Cambridge Biomedical Campus, Cambridge, UK. [28]Moorfields Eye Hospital NHS Foundation Trust, London, UK. [29]UCL Institute of Ophthalmology, University College London, London, UK. [30]Department of Haematology, University of Cambridge, Cambridge Biomedical Campus, Cambridge, UK. [31]MRC Biostatistics Unit, Cambridge Institute of Public Health, University of Cambridge, Cambridge, UK. [32]MRC Molecular Haematology Unit, Weatherall Institute of Molecular Medicine, University of Oxford, Oxford, UK. [33]NIHR Oxford Biomedical Research Centre, Oxford University Hospitals Trust, Oxford, UK. [34]Center for Clinical Transfusion Medicine, University Hospital of Tübingen, Tübingen, Germany. [35]The Katharine Dormandy Haemophilia Centre and Thrombosis Unit, Royal Free London NHS Foundation Trust, London, UK. [36]University College London, London, UK. [37]Department of Clinical Genetics, Leicester Royal Infirmary, University Hospitals of Leicester, Leicester, UK. [38]University of Leicester, Leicester, UK. [39]The Nuffield Department of Clinical Neurosciences, University of Oxford, John Radcliffe Hospital, Oxford, UK. [40]Illumina Limited, Chesterford Research Park, Little Chesterford, Nr Saffron Walden, UK. [41]Bristol Renal, University of Bristol, Bristol, UK. [42]Haematology Department, Royal Victoria Infirmary, The Newcastle upon Tyne Hospitals NHS Foundation Trust, Newcastle upon Tyne, UK. [43]Department of Medicine, School of Clinical Medicine, University of Cambridge, Cambridge Biomedical Campus, Cambridge, UK. [44]Department of Pulmonary Medicine, VU University Medical Centre, Amsterdam, The Netherlands. [45]Southampton General Hospital, University Hospital Southampton NHS Foundation Trust, Southampton, UK. [46]MRC Social, Genetic & Developmental Psychiatry Centre, Institute of Psychiatry, Psychology & Neuroscience, King's College London, London, UK. [47]NIHR Biomedical Research Centre for Mental Health, Maudsley Hospital, London, UK. [48]Newcastle University, Newcastle upon Tyne, UK. [49]Newcastle upon Tyne Hospitals NHS Foundation Trust, Newcastle upon Tyne, UK. [50]Department of Clinical Genetics, Royal Devon & Exeter Hospital, Royal Devon and Exeter NHS Foundation Trust, Exeter, UK. [51]Department of Immunology, Leicester Royal Infirmary, Leicester, UK. [52]National Heart and Lung Institute, Imperial College London, Royal Brompton Hospital, London, UK. [53]Royal Brompton Hospital, Royal Brompton and Harefield NHS Foundation Trust, London, UK. [54]Royal Free London NHS Foundation Trust, London, UK. [55]King's College London, London, UK. [56]Guy's and St Thomas' Hospital, Guy's and St Thomas' NHS Foundation Trust, London, UK. [57]King's College Hospital NHS Foundation Trust, London, UK. [58]High Performance Computing Service, University of Cambridge, Cambridge, UK. [59]Epidemiology and Biostatistics, Imperial College London, London, UK. [60]Imperial College Healthcare NHS Trust, London, UK. [61]Department of Clinical Neurosciences, School of Clinical Medicine, University of Cambridge, Cambridge Biomedical Campus, Cambridge, UK. [62]Medical Research Council Mitochondrial Biology Unit, Cambridge Biomedical Campus, Cambridge, UK. [63]Children's Renal and Urology Unit, Nottingham Children's Hospital, QMC, Nottingham University Hospitals NHS Trust, Nottingham, UK. [64]Golden Jubilee National Hospital, Glasgow, UK. [65]West Midlands Regional Genetics Service, Birmingham Women's and Children's NHS Foundation Trust, Birmingham, UK. [66]The Royal London Hospital, Barts Health NHS Foundation Trust, London, UK. [67]The Arthur Bloom Haemophilia Centre, University Hospital of Wales, Cardiff, UK. [68]Sheffield Pulmonary Vascular Disease Unit, Royal Hallamshire Hospital NHS Foundation Trust, Sheffield, UK. [69]MRC London Institute of Medical Sciences, Imperial College London, London, UK. [70]National Heart Research Institute Singapore, National Heart Centre Singapore, Singapore, Singapore. [71]Division of Cardiovascular & Metabolic Disorders, Duke- National University of Singapore, Singapore, Singapore. [72]Imperial College Renal and Transplant Centre, Hammersmith Hospital, Imperial College Healthcare NHS Trust, London, UK. [73]Department of Medicine, Imperial College London, London, UK. [74]National Pulmonary Hypertension Service (Newcastle), The Newcastle upon Tyne Hospitals NHS Foundation Trust, Newcastle upon Tyne, UK. [75]Institute of Cellular Medicine, Faculty of Medical Sciences, Newcastle University, Newcastle upon Tyne, UK. [76]Oxford Haemophilia and Thrombosis Centre, The Churchill Hospital, Oxford University Hospitals NHS Trust, Oxford, UK. [77]Department of Molecular Medicine, General Biology, and Medical Genetics Unit, University of Pavia, Pavia, Italy. [78]Department of Cardiovascular Medicine, Radcliffe Department of Medicine, University of Oxford, Oxford, UK. [79]Oxford University Hospitals NHS Foundation Trust, Oxford, UK. [80]Stroke Research Group, Department of Clinical Neurosciences, University of Cambridge, Cambridge Biomedical Campus, Cambridge, UK. [81]Nottingham University Hospitals NHS Trust, Nottingham, UK. [82]The Roald Dahl Haemostasis and Thrombosis Centre, The Royal Liverpool Hospital, Liverpool, UK. [83]Regional Immunology Service, Kelvin Building, Royal Victoria Hospital, Belfast, UK. [84]Sheffield Teaching Hospitals NHS Foundation Trust, Sheffield, UK. [85]Pathology and Laboratory Medicine, University of Western Australia, Crawley, WA, Australia. [86]Haemophilia Centre, Kent & Canterbury Hospital, East Kent Hospitals University Foundation Trust, Canterbury, UK. [87]Manchester Centre for Genomic Medicine, Saint Mary's Hospital, Manchester, UK. [88]Salisbury District Hospital, Salisbury NHS Foundation Trust, Salisbury, UK. [89]Departement de genetique, Hopital Pitie-Salpetriere, Paris, France. [90]Service d'Hématologie Biologique, Hôpital d'enfants Armand Trousseau, Paris, France, Paris, France. [91]Inserm U1170, Villejuif, France. [92]Assistance Publique-Hôpitaux de Paris, Département d'Hématologie, Hôpital Armand Trousseau, Paris, France. [93]Department of Paediatrics, School of Clinical Medicine, University of Cambridge, Cambridge Biomedical Campus, Cambridge, UK. [94]Department of Cardiovascular Sciences, Center for Molecular and Vascular Biology, University of Leuven, Leuven, Belgium. [95]University of Giessen and Marburg Lung Center (UGMLC), Giessen, Germany. [96]Division of Cardiology, Fondazione IRCCS Policlinico S. Matteo, Pavia, Italy. [97]UCL Great Ormond Street Institute of Child Health, London, UK. [98]Universite Paris-Sud, Le Kremlin-Bicêtre, France. [99]Service de Pneumologie, DHU Thorax Innovation, Hôpital Bicêtre, Le Kremlin-Bicêtre, France. [100]INSERM U999, LabEx LERMIT, Centre Chirurgical Marie Lannelongue, Le Plessis Robinson, France. [101]University Hospitals of North Midlands NHS Trust, Stoke-on-Trent, UK. [102]East Yorkshire Regional Adult Immunology and Allergy Unit, Hull Royal Infirmary, Hull & East Yorkshire Hospitals NHS Trust, Hull, UK. [103]Department of Clinical Genetics, Liverpool Women's NHS Foundation, Liverpool, UK. [104]Institute for Immunology and Transfusion Medicine, University of Greifswald, Greifswald, Germany. [105]Section of Internal and Cardiovascular Medicine, University of Perugia, Perugia, Italy. [106]Mitochondrial Research Group, Institute of Genetic Medicine, Newcastle University, Newcastle upon Tyne, UK. [107]Institute of Genetic Medicine, Newcastle University, Newcastle upon Tyne, UK. [108]Barts Health NHS Foundation Trust, London, UK. [109]Birmingham Heartlands Hospital, Heart of England NHS Foundation Trust, Birmingham, UK. [110]Department of Neurology, Sheffield Teaching Hospitals NHS Foundation Trust, Sheffield, UK. [111]Wellcome Trust Centre for Human Genetics, University of Oxford, Oxford, UK. [112]Department of Neurology, Leeds Teaching Hospital NHS Trust, Leeds, UK. [113]Epsom & St Helier University Hospitals NHS Trust, London, UK. [114]Northern Genetics Service, The Newcastle upon Tyne Hospitals NHS Foundation Trust, International Centre for Life, Newcastle upon Tyne, UK. [115]Wellcome Centre for Mitochondrial Research, Institute of Genetic Medicine, Newcastle University, Newcastle upon Tyne, UK. [116]John Walton Muscular Dystrophy Research Centre, Institute of Genetic Medicine, Newcastle University, Newcastle upon Tyne, UK. [117]Wellcome Trust Sanger Institute, Hinxton, Cambridge, UK. [118]Department of Clinical Genetics, Guy's and St Thomas' NHS Foundation Trust, London, UK. [119]Department of Paediatric Nephrology, Great North Children's Hospital, Newcastle upon Tyne Hospitals NHS Foundation Trust, Newcastle upon Tyne, UK. [120]University Hospital of Wales, Cardiff, UK. [121]Cardiff & Vale University LHB, Cardiff, UK. [122]Addenbrookes Hospital, Cambridge University Hospitals NHS Foundation Trust, Cambridge, UK. [123]University College London Hospitals NHS Foundation Trust, London, UK. [124]Ludwig Boltzmann Institute for Lung Vascular Research, Graz, Austria. [125]Department of Internal Medicine, Division of Pulmonology, Medical University of Graz, Graz, Austria. [126]Division of Transplantation Immunology and Mucosal Biology, Department of Experimental Immunobiology, Faculty of Life

Sciences and Medicine, King's College London, London, UK. [127]Department of Paediatric Nephrology, Evelina London Children's Hospital, Guy's & St Thomas' NHS Foundation Trust, London, UK. [128]Department of Pediatric Hematology, Immunology, Rheumatology and Infectious Diseases, Emma Children's Hospital, Academic Medical Center (AMC), University of Amsterdam, Amsterdam, The Netherlands. [129]Department of Clinical Genetics, Academic Medical Center (AMC), University of Amsterdam, Amsterdam, The Netherlands. [130]Molecular Neurosciences, Developmental Neurosciences, UCL Great Ormond Street Institute of Child Health, London, UK. [131]Department of Neurology, Great Ormond Street Hospital for Children NHS Foundation Trust, London, UK. [132]Department of Haematology, Hammersmith Hospital, Imperial College Healthcare NHS Trust, London, UK. [133]Department of Haematology, Imperial College London, London, UK. [134]Division of Hematology, The Children's Hospital of Philadelphia, Philadelphia, PA, USA. [135]Department of Pediatrics, Perelman School of Medicine at the University of Pennsylvania, Philadelphia, PA, USA. [136]Department of Infection, Immunity & Cardiovascular Disease, University of Sheffield, Sheffield, UK. [137]Royal United Hospitals Bath NHS Foundation Trust, Bath, UK. [138]Department of Haematology, Guy's and St Thomas' NHS Foundation Trust, London, UK. [139]Cambridge NIHR Biomedical Research Centre, Cambridge Biomedical Campus, Cambridge, UK. [140]Haemophilia, Haemostasis and Thrombosis Centre, Hampshire Hospitals NHS Foundation Trust, Basingstoke, UK. [141]Faculty of Medical and Human Sciences, Centre for Endocrinology and Diabetes, Institute of Human Development, University of Manchester, Manchester, UK. [142]Department of Clinical Neurophysiology, Manchester Royal Infirmary, Central Manchester University Hospitals National Health Service Foundation Trust, Manchester Academic Health Science Centre, Manchester, UK. [143]National Institute for Health Research/Wellcome Trust Clinical Research Facility, Manchester, UK. [144]Department of Haematology, Great Ormond Street Hospital for Children NHS Foundation Trust, London, UK. [145]The National Hospital for Neurology and Neurosurgery, University College London Hospitals NHS Foundation Trust, London, UK. [146]Royal Hospital for Children, NHS Greater Glasgow and Clyde, Glasgow, UK. [147]Oxford Centre for Diabetes, Endocrinology and Metabolism, University of Oxford, Churchill Hospital, Oxford, UK. [148]Centre for Haematology, Department of Medicine, Hammersmith Hospital, Imperial College Healthcare NHS Trust, London, UK. [149]Department of Clinical Genetics, Addenbrookes Hospital, Cambridge University Hospitals NHS Foundation Trust, Cambridge, UK. [150]Institute of Neuroscience and Psychology, University of Glasgow, Glasgow, UK. [151]School of Cellular and Molecular Medicine, University of Bristol, Bristol, UK. [152]University Hospitals Bristol NHS Foundation Trust, Bristol, UK. [153]East Anglian Regional Genetics Service, Cambridge University Hospitals NHS Foundation Trust, Cambridge, UK. [154]Royal Papworth Hospital NHS Foundation Trust, Cambridge, UK. [155]Pain Research, Department of Surgery and Cancer, Faculty of Medicine, Imperial College London, London, UK. [156]Chelsea and Westminster Hospital NHS Foundation Trust, London, UK. [157]Department of Paediatrics, Weatherall Institute of Molecular Medicine, University of Oxford, Oxford, UK. [158]UCL Centre for Nephrology, University College London, London, UK. [159]Departments of Cardiovascular Sciences and NIHR Leicester Cardiovascular Biomedical Research Unit, University of Leicester, Leicester, UK. [160]The Leeds Teaching Hospitals NHS Trust, Leeds, UK. [161]Beth Israel Deaconess Medical Centre and Harvard Medical School, Boston, MA, USA. [162]Department of Clinical Genetics, Nottingham University Hospitals NHS Trust, Nottingham, UK. [163]Scunthorpe General Hospital, Northern Lincolnshire and Goole NHS Foundation Trust, Scunthorpe, UK. [164]Clinical Genetics Institute, Kaplan Medical Center, Rehovot, Israel. [165]Genetics and Molecular Medicine, King's College London, London, UK. [166]Aarhus University Hospital, Aarhus, Denmark. [167]Cambridge Institute for Medical Research, University of Cambridge, Cambridge Biomedical Campus, Cambridge, UK. [168]Department of Clinical Genetics, St George's University Hospitals NHS Foundation Trust, London, UK. [169]Glasgow Royal Infirmary, NHS Greater Glasgow and Clyde, Glasgow, UK. [170]Gartnavel General Hospital, NHS Greater Glasgow and Clyde, Glasgow, UK. [171]Addenbrooke's Treatment Centre, Addenbrooke's Hospital, Cambridge University Hospitals NHS Foundation Trust, Cambridge, UK. [172]Department of Molecular Neuroscience, UCL Institute of Neurology, London, UK. [173]UCL Genetics Institute, London, UK. [174]Frimley Park Hospital, NHS Frimley Health Foundation Trust, Camberley, UK.

## Genomics England Research Consortium Collaborators

J. C. Ambrose[7], P. Arumugam[7], R. Bevers[7], M. Bleda[7], F. Boardman-Pretty[7,175], C. R. Boustred[7], H. Brittain[7], M. A. Brown[7], M. J. Caulfield[7,175], G. C. Chan[7], T. Fowler[7], A. Giess[7], A. Hamblin[7], S. Henderson[7,175], T. J. P. Hubbard[7], R. Jackson[7], L. J. Jones[7,175], D. Kasperaviciute[7,175], M. Kayikci[7], A. Kousathanas[7], L. Lahnstein[7], S. E. A. Leigh[7], I. U. S. Leong[7], F. J. Lopez[7], F. Maleady-Crowe[7], M. McEntagart[7], F. Minneci[7], L. Moutsianas[7,175], M. Mueller[7,175], N. Murugaesu[7], A. C. Need[7,175], P. O'Donovan[7], C. A. Odhams[7], C. Patch[7,175], D. Perez-Gil[7], M. B. Pereira[7], J. Pullinger[7], T. Rahim[7], A. Rendon[7], T. Rogers[7], K. Savage[7], K. Sawant[7], R. H. Scott[7], A. Siddiq[7], A. Sieghart[7], S. C. Smith[7], A. Sosinsky[7,175], A. Stuckey[7], M. Tanguy[7], A. L. Taylor Tavares[7], E. R. A. Thomas[7,175], S. R. Thompson[7], A. Tucci[7,175], M. J. Welland[7], E. Williams[7], K. Witkowska[7,175] & S. M. Wood[7,175]

[175]William Harvey Research Institute, Queen Mary University of London, London EC1M 6BQ, UK.

