## [Peer Review File · Nature Communications]

Genome-Wide Association study of Intrahepatic Cholestasis of Pregnancy Identifies Several Susceptibility Loci in Liver-Specific Regulatory ElementsReviewers' Comments:

Reviewer #1:

Remarks to the Author:

Very impressive dataset for an understudied clinical problem. I read it as a clinician, not a statistician.

As I read it I had the following thoughts

- a) can the authors not use the term liver function tests- serum liver tests is more appropriate;
- b) the disorder is pregnancy specific, in the bigger context of cholestasis (genetic, drug induced, autoimmune); but the authors unless I missed it have not shown why the findings are pregnancy specific- are the statistical associations related to estrogen/progesterone signalling effects on these loci i.e. why are these regulatory associations so pregnancy specific; are there any experimental models/cell line studies they can perform? These changes must occur in men as well?
- c) I really appreciate a good study design figure- it highlights the flow of the analysis and all the cohorts
- d) I really appreciate reading a paragraph with clear a priori definitions for significance at each stage and for choosing gene areas for further evaluation- it is always a concern that non-statistical confounders are introduced, so better to see that loci evaluation is per a stats plan;
- d) I am struck that this disease is perhaps really an environmental one by definition. The environment in which these subtle variations are pregnancy but beyond that do the authors agree that there is variation in risk geographically? Beyond pregnancy hormones, any other clues to environmental contributors from the data? Certain geographic populations have more frequent OC-how can that be assessed?
- e) can you establish a genetic risk score? Any pathway analysis? Are there gene-gene interactions/gene-dose effects?
- f) What about drug repurposing analyses? Are any pathways modifiable by existing drugs?
- g) Limitations: with this data can the authors estimate the genetic contribution of OC again? Is the study still underpowered? What is the most informative OC cohort to study- familial, high bile acids, recurrent?
- h) Do the authors know if any of the cohort developed cholestasis in any non-pregnancy setting?
- i) liver biochemical tests indeed also have genetic contributions e.g. for ALP, ALT. There is recent NAFLD data as well. Anyway to assess this?

Reviewer #2:

Remarks to the Author:

The manuscript by Dixon and co-authors details a genome-wide association scan for intra-hepatic cholestasis of pregnancy, and the subsequent bioinformatic and lab-based follow-up of associated loci. This is a paper that will be of great interest to the study of this and other liver diseases given the findings of common variants around some known and also novel bile acid metabolism genes. While some protein disrupting rare variants have previously been shown to be associated with increased ICP risk in some genes in these pathways (particularly ABCB4), bile acid metabolism genes in particular have been the subject of multiple candidate studies for this condition (that have, frustratingly, found very little), and the association of common variants in this pathway will undoubtedly open up further avenues of future research. The research presented here is nicely laid out, from the GWAS through to the follow-up, and follows a nice model for such studies in incorporating functional evidence to support GWAS findings.

My major issue with this paper is with the size, make-up and description of both the discovery and

replication samples, which is too lacking in detail for clarity around the initial results. Both the discovery and replication sets contain individuals from multiple countries, and it's not clear how appropriate the controls are for the cases. The discovery sample has an exceptionally small number of cases (287) from a GWAS perspective. The authors have presumably attempted to boost power by including only severe ICP cases (although this is not explicitly stated), however, these samples come from the UK, Australia, Argentina and Sweden. There are no details (particularly the N) here of the source of the samples, only a reference to the larger Bioresource they were taken from). While controls come from the same Bioresource, the composition of the control group is not described at all, nor was there any mention of matching controls by location/ancestry. I'm concerned about this because comparing 287 cases to over 13,000 controls yielded one locus reaching genome-wide significance ($P < 5 \times 10^{-8}$), and five reaching suggestive significance ($P < 5 \times 10^{-7}$). While none of these are the result of signal from loner SNPs (which would be concerning), and appear around biologically plausible genes, the large odds ratios do raise a red flag for me – these are not what you would expect to see from a heterogeneous trait (where typical GWAS effect sizes tend to be much closer to 1). Have you done any power calculations to see what sample size would you need to see effects that are really that large? It would be easy to tell a story around the involvement of only a few liver-specific genes in a highly-selected population (which I agree might be plausible), but I'm concerned the extent to which the ancestry of the samples has influenced your results, which hasn't been explored or even mentioned here. You've included 5 principal components as covariates in your analysis, but is this enough? As a reader I have no idea of the breakdown of this sample in terms of N from each population, or that there is admixture as this isn't mentioned in the main manuscript, and it's not until line 71 of the supplementary that you mention all individuals were analysed as one sample in your discovery analysis. This isn't typical, or justified in your manuscript. For any GWAS at the minimum I would expect to see a better description in the main text particularly if multiple ancestries are involved, a supplementary Table outlining your sample, and even a forest plot of effect sizes per sample/ancestry group.

You also mention relatedness in your sample, finding a subset of 10,516 unrelated individuals, but your discovery sample is larger than this, so presumably includes relatives? In which case you would be better to use a program that can handle related individuals (e.g. raremetalworker would be a simple switch given you have a small sample overall, that is already formatted for plink).

The replication sample is also not well described, and again only in the supplementary text. The description of this sample is left to other references, but I had to search through three (ref 2 in the supp, then another paper cited in that reference, and ref 41), to try to determine which samples were included in this set, which indicate the majority of replication cases were recruited in Sweden (although that paper indicates more samples than you used here), and the controls in Ireland? You need to justify this. You also give allele frequencies for non-Finnish samples (note this is miss-spelt as 'non-Finish' in both the main and supplementary text), does that mean you have Finnish samples in your replication set? These aren't mentioned in your replication set description. Given you 'replicate' 5 of your loci with a predominantly Swedish sample makes me also wonder if your discovery sample is predominantly Swedish, in which case are you picking up ancestry as well as possible ICP case-control differences?

I also suggest you meta-analyse your discovery and replication results, and include the meta-analysis results (as well as allele frequencies for your discovery set) in Table 1. Eye-balling these it looks like you will get the same loci surpassing genome-wide significance, giving you more confidence in your results than a replication set surpassing a not very stringent multiple testing threshold.

****Minor issues**

Please provide more explanation for your table headings (both main text and supplementary tables),

rather than just program output.

On line 194 of the main text you indicate you have found 'the key' genes contributing to the aetiopathogenesis of ICP. I would take out at least 'the', and preferably 'key' as well – your sample size just isn't big enough to claim you've found the key genes underlying ICP susceptibility. Focusing on the liver specificity, which is backed up by your functional work, would be more appropriate here.

Reviewer #3:

Remarks to the Author:

This is a very well-written, suitably concise report of an interesting study that says much about the genetic basis of ICP. Of particular interest is that common variants confirmed in this study implicate comparable biological processes to rare variants/mutations known to be associated with the condition. The authors are leaders in the field.

It is not clear to me whether the authors set out to explore common variation in ICP, or a common variant GWAS was undertaken because whole genome sequence data were available but the sample size meant that only common variants with strong effects could be reliably investigated. Clarification would be appropriate.

Population stratification might be an issue. The PCA plots suggest that not all samples cluster, but the supplementary text suggests that samples which didn't cluster were included anyway. The authors are evidently confident that this was acceptable. More detail about these considerations might be appropriate in the supplementary text. For example, are they satisfied that they have fully accounted for population stratification? Why do they feel that outliers improve rather than confound the analysis?

In validation, individuals and SNPs with >50% missingness seems lenient.

In validation, five of the association signals identified in discovery demonstrated significant evidence of association in validation (main text, line 115) – and all but the FADS2 locus remained significant after correcting for multiple testing (main text, line 116). The authors state in the supplementary text that, "The statistical significance threshold correcting for examining six independent loci was $p < 0.0083$ " (supplementary text, line 94). Therefore, my interpretation is that four of the association signals identified in discovery demonstrated significant evidence of association in validation according to the authors own definition of significance ($p < 0.0083$). The authors subsequently discuss FADS2 as though it has been validated. If the authors prefer to define validation as $P < 0.05$, they should be clear about this.

The functional annotation is contemporary and informative. The luciferase reporter assay supports the functional annotation.

Very minor comments are that the abstract cites references, which probably is not necessary, and there minor typographical errors in the supplementary text which should be corrected (e.g. lines 78 & 79).

Overall, I consider this an important study. It may, however, be more suitable for a specialty journal.

Reviewer #1 (Remarks to the Author):

Very impressive dataset for an understudied clinical problem. I read it as a clinician, not a statistician.

As I read it I had the following thoughts

a) can the authors not use the term liver function tests- serum liver tests is more appropriate;

We have used the term 'serum liver tests' in the revised version of the manuscript, page 2 line 51 and page 3 line 69.

b) the disorder is pregnancy specific, in the bigger context of cholestasis (genetic, drug induced, autoimmune); but the authors unless I missed it have not shown why the findings are pregnancy specific- are the statistical associations related to estrogen/progesterone signalling effects on these loci i.e. why are these regulatory associations so pregnancy specific; are there any experimental models/cell line studies they can perform? These changes must occur in men as well?

We have added the following text to the discussion to address this comment:

"In the context of pregnancy, raised gestational hormones cause reduced FXR-mediated induction of hepatic Bsep, Shp and Mdr3 (PMID 22961653), and enterocyte Shp and Fgf15/19 (PMID 32127609) thereby causing abnormal bile acid homeostasis. This is likely to exacerbate susceptibility to hypercholanemia in genetically predisposed women with the variants reported in this manuscript." page 12, lines 293-7

c) I really appreciate a good study design figure- it highlights the flow of the analysis and all the cohorts

Thank you for this suggestion. A flow chart summarising the study design including the various cohorts and their sizes has been included in the revised manuscript as Supplementary Figure 1.

d) I really appreciate reading a paragraph with clear a priori definitions for significance at each stage and for choosing gene areas for further evaluation- it is always a concern that non-statistical confounders are introduced, so better to see that loci evaluation is per a stats plan

In the revised version of the manuscript, we have clearly stated the *a priori* statistical threshold (page 5, line 126) for considering loci for further evaluation which is the community accepted standard of $p < 5 \times 10^{-8}$. Statistical significance thresholds for other analyses have been defined accordingly in the text.

d) I am struck that this disease is perhaps really an environmental one by definition. The environment in which these subtle variations are pregnancy but beyond that do the authors agree that there is variation in risk geographically? Beyond pregnancy hormones, any other

clues to environmental contributors from the data? Certain geographic populations have more frequent OC-how can that be assessed?

We believe that the current manuscript clearly shows that the etiology of ICP has a genetic component; as the endocrine changes of normal pregnancy cause hypercholanemia, we believe the condition is caused by a combination of genetic susceptibility and the cholestatic effect of gestational hormonal alterations. In addition to this, the prevalence of ICP varies in different geographic regions as the reviewer has said. This could be explained by genetic variation or by environmental influences. We have therefore added the following text to the manuscript:

“A variable geographical prevalence has also been observed with this disease. This may be explained by genetic variation in women of different ancestry, or could be the result of altered environmental factors, e.g. vitamin D or selenium deficiency (PMID 10782901, 20959506).”

e) can you establish a genetic risk score? Any pathway analysis? Are there gene-gene interactions/gene-dose effects?

- 1. We can generate a genetic risk score; however, by definition this will show an enrichment in ICP compared with controls and not provide additional useful information unless combined with extra phenotype data that is not available presently.**
- 2. However, given the genetic overlap with cholelithiasis we observed, we have now used genetic risk scores to show that the disorders have distinct but overlapping genetic risk factors.**
- 3. In the revised version of the manuscript, we have included unbiased pathway analyses that demonstrate that the loci identified contain genes showing preferential expression in the liver and highlighting pathophysiologically relevant biochemical pathways.**
- 4. We have tested for epistasis between each of the 55 pairs of the lead variants at the now 11 genome-wide significant loci in the two UK cohorts in the revised manuscript. There is no evidence of interactions. This has been described in the revision.**

f) What about drug repurposing analyses? Are any pathways modifiable by existing drugs?

We thank the reviewer for this interesting comment. It is beyond the scope of this paper to study this in detail.

g) Limitations: with this data can the authors estimate the genetic contribution of OC again? Is the study still underpowered? What is the most informative OC cohort to study- familial, high bile acids, recurrent?

- 1. Accurately estimating the genetic contribution of the identified genetic risk loci to the phenotypic variance in ICP requires the population prevalence to be known. Given the**

transient nature of the phenotype in pregnancy it is not possible to calculate the this in the standard manner.

2. Power calculations have been performed and incorporated into the revised manuscript. Given the number of cases and controls, the GWAS meta-analysis was powered (at 80%) to detect associations with odds ratio (OR) > 1.3 at a minor allele frequency (MAF) of ≥ 0.5 and OR > 1.7 at MAF ≥ 0.05 . The power at varying OR and MAF is shown in Supplementary Figure 4. A greater number of cases would be required to enable associations with weaker effects to be detected at genome-wide significance.
3. The most informative cohort to study is that which gives us the largest sample size. This is achieved by combining all available GWAS data from ICP patients available yielding the largest sample size studied in a meta-analysis to date.

h) Do the authors know if any of the cohort developed cholestasis in any non-pregnancy setting?

We suspect a small proportion of cases in the NIHR-RD cohort developed oral contraceptive/drug-induced cholestasis, based on their reporting of itching. The details of these findings have been added to the text:

“For women that used hormonal contraception, 15/254 (5.9%) reported itching when used and 39/256 reported cyclical itching. Additionally, 15/247 patients (6.1%) reported drug-induced itching (other than for contraception)”, page 4 line 98.

This deep phenotyping data is not available for the other cohorts, but we believe the proportions would be similar.

i) liver biochemical tests indeed also have genetic contributions e.g. for ALP, ALT. There is recent NAFLD data as well. Anyway to assess this?

We have checked for overlap between the loci identified in this study and all previously published associations as per the GWAS Catalogue. Some of the loci identified are indeed associated with serum liver tests, NAFLD, cholestatic disorders and lipids, amongst other traits. These results have been included in the revised manuscript (Extended Data 2 and Supplementary Figure 6).

Reviewer #2 (Remarks to the Author):

The manuscript by Dixon and co-authors details a genome-wide association scan for intra-hepatic cholestasis of pregnancy, and the subsequent bioinformatic and lab-based follow-up of associated loci. This is a paper that will be of great interest to the study of this and other liver diseases given the findings of common variants around some known and also novel bile acid metabolism genes. While some protein disrupting rare variants have previously been shown to be associated with increased ICP risk in some genes in these pathways (particularly ABCB4), bile acid metabolism genes in particular have been the subject of multiple candidate studies for this condition (that have, frustratingly, found very little), and

the association of common variants in this pathway will undoubtedly open up further avenues of future research. The research presented here is nicely laid out, from the GWAS through to the follow-up, and follows a nice model for such studies in incorporating functional evidence to support GWAS findings.

We thank the reviewer for these supportive comments.

My major issue with this paper is with the size, make-up and description of both the discovery and replication samples, which is too lacking in detail for clarity around the initial results.

The revised manuscript is significantly improved and addresses all the issues and queries raised by this reviewer regarding the discovery GWAS and replication analyses. In the initial submission we performed a multi-ancestry GWAS with 287 cases and sought replication of suggestive and significant loci in a cohort of 606 cases without genome-wide data for genetic ancestry assessment. In the revised version of the manuscript, we have performed GWAS and meta-analysis in two largely UK based cohorts of 398 ICP cases all of genetically confirmed European ancestry. The resulting data were meta-analysed with GWAS summary statistics from the publicly available FinnGen data with 740 ICP cases. Descriptions regarding each of the cohorts have been included in the revised manuscript.

Both the discovery and replication sets contain individuals from multiple countries, and it's not clear how appropriate the controls are for the cases. The discovery sample has an exceptionally small number of cases (287) from a GWAS perspective. The authors have presumably attempted to boost power by including only severe ICP cases (although this is not explicitly stated), however, these samples come from the UK, Australia, Argentina and Sweden.

In the revised manuscript we have restricted all analyses to individuals of genetically confirmed European ancestry. The identification of this subset of individuals has been undertaken using community accepted standard methodology. We are confident that both the cases and controls are drawn from a genetically homogenous population. This is reflected by the low genomic inflation factors (λ s) indicating no significant population stratification. Importantly we have now meta-analysed the data from the UK with Finnish GWAS data (across a total of 1,138 cases) which replicates the associations in the UK data and identifies additional novel genome-wide significant findings. Whilst we acknowledge that the number of cases is still relatively small from a GWAS perspective post-hoc power calculations demonstrate that we are sufficiently powered to identify moderately sized associations (see above) and we have clearly done so.

There are no details (particularly the N) here of the source of the samples, only a reference to the larger Bioresource they were taken from). While controls come from the same Bioresource, the composition of the control group is not described at all, nor was there any mention of matching controls by location/ancestry.

As described above, in the revised version of the manuscript, we have restricted our analyses to individuals of genetically confirmed European ancestry. In the Supplementary Information we have included additional information on the case/control composition of each of the cohorts included in the revised manuscript.

I'm concerned about this because comparing 287 cases to over 13,000 controls yielded one locus reaching genome-wide significance ($P < 5 \times 10^{-8}$), and five reaching suggestive significance ($P < 5 \times 10^{-7}$). While none of these are the result of signal from loner SNPs (which would be concerning), and appear around biologically plausible genes, the large odds ratios do raise a red flag for me – these are not what you would expect to see from a heterogeneous trait (where typical GWAS effect sizes tend to be much closer to 1). Have you done any power calculations to see what sample size would you need to see effects that are really that large?

As mentioned above, we have undertaken post-hoc power calculations that have demonstrated we are well powered to detect associations with effect sizes and allele frequencies seen for the genome-wide significant variants. Furthermore, the association signals at the loci achieving genome-wide significance are supported by strong underlying data with increasing significance, they are biologically plausible and, importantly, they replicate and withstand meta-analysis with an independent cohort. We are therefore confident that these reflect genuine statistical and biological associations.

It would be easy to tell a story around the involvement of only a few liver-specific genes in a highly-selected population (which I agree might be plausible), but I'm concerned the extent to which the ancestry of the samples has influenced your results, which hasn't been explored or even mentioned here.

In the revised version of the manuscript, the analyses have been restricted to individuals of European ancestry and this point is therefore no longer applicable.

You've included 5 principal components as covariates in your analysis, but is this enough?

In the revised version of the manuscript, we have included ten principal components in the GWAS. This is standard in the field. The low genomic inflation indicates that population stratification is not confounding the results.

As a reader I have no idea of the breakdown of this sample in terms of N from each population, or that there is admixture as this isn't mentioned in the main manuscript, and it's not until line 71 of the supplementary that you mention all individuals were analysed as one sample in your discovery analysis. This isn't typical, or justified in your manuscript. For any GWAS at the minimum I would expect to see a better description in the main text particularly if multiple ancestries are involved, a supplementary Table outlining your sample, and even a forest plot of effect sizes per sample/ancestry group.

The issues raised here are no longer applicable given the way in which the analysis has been done in the revised manuscript.

You also mention relatedness in your sample, finding a subset of 10,516 unrelated individuals, but your discovery sample is larger than this, so presumably includes relatives? In which case you would be better to use a program that can handle related individuals (e.g. raremetalworker would be a simple switch given you have a small sample overall, that is already formatted for plink).

The original cohorts from which the study subjects (both cases and controls) were drawn include some related individuals; however, for each cohort filtering for relatedness was performed and only unrelated individuals were included. This is described in the methods.

The replication sample is also not well described, and again only in the supplementary text. The description of this sample is left to other references, but I had to search through three (ref 2 in the supp, then another paper cited in that reference, and ref 41), to try to determine which samples were included in this set, which indicate the majority of replication cases were recruited in Sweden (although that paper indicates more samples than you used here), and the controls in Ireland? You need to justify this.

In the revised version of the manuscript, we have not utilised the previously described replication data. As we do not have genome-wide SNP data from these individuals (or sufficient DNA available remaining to generate such data) it is not possible to definitively assess genetic ancestry. We have therefore elected to not utilise these data. All individuals included in the analyses in the revised manuscript are of genetically defined European ancestry.

You also give allele frequencies for non-Finnish samples (note this is miss-spelt as 'non-Finish' in both the main and supplementary text), does that mean you have Finnish samples in your replication set? These aren't mentioned in your replication set description. Given you 'replicate' 5 of your loci with a predominantly Swedish sample makes me also wonder if your discovery sample is predominantly Swedish, in which case are you picking up ancestry as well as possible ICP case-control differences?

We have done our utmost to ensure there are no such spelling mistakes in the revised version of the manuscript. Our UK analysis is based on European individuals not enriched for Finnish individuals. These results are now meta-analysed with data from Finnish individuals available from FinnGen. Of the 11 loci identified as achieving genome-wide significance in the combined meta-analysis, 4 were significant in the separate much smaller UK dataset. However, at all the 7 additional loci, there was at least a nominally significant association with ICP in the UK data. This indicates that these effects are not driven by differences in ancestry.

I also suggest you meta-analyse your discovery and replication results, and include the meta-analysis results (as well as allele frequencies for your discovery set) in Table 1. Eye-balling these it looks like you will get the same loci surpassing genome-wide significance, giving you more confidence in your results than a replication set surpassing a not very stringent multiple testing threshold.

We thank the reviewer for this suggestion. In the revised version of the manuscript, we have performed meta-analyses resulting in the identification of 11 loci achieving genome-wide significance.

****Minor issues**

Please provide more explanation for your table headings (both main text and supplementary tables), rather than just program output.

We have expanded our explanation of the table headings in the table description.

On line 194 of the main text you indicate you have found ‘the key’ genes contributing to the aetiopathogenesis of ICP. I would take out at least ‘the’, and preferably ‘key’ as well – your sample size just isn’t big enough to claim you’ve found the key genes underlying ICP susceptibility. Focusing on the liver specificity, which is backed up by your functional work, would be more appropriate here.

The word ‘the’ has been removed and the text now states, “identifying key contributing genes”, page 14 line 340. Given the extensive new amount of data incorporated into the meta-analysis we present in the revised version of the manuscript, we would like to argue that we have indeed identified some genes that are key to define the genetic architecture of ICP. This is further supported by the observation that the majority of risk loci are also associated with ICP-relevant traits, such as gallstone disease (Supplementary Figure 8).

Reviewer #3 (Remarks to the Author):

This is a very well-written, suitably concise report of an interesting study that says much about the genetic basis of ICP. Of particular interest is that common variants confirmed in this study implicate comparable biological processes to rare variants/mutations known to be associated with the condition. The authors are leaders in the field.

We thank the reviewer for these comments.

It is not clear to me whether the authors set out to explore common variation in ICP, or a common variant GWAS was undertaken because whole genome sequence data were available but the sample size meant that only common variants with strong effects could be reliably investigated. Clarification would be appropriate.

The objective of this study was to explore the role of common (>1%) genome-wide variation in ICP. This was both because (i) it has never been done before in a systematic manner and (ii) the modest number of cases available limited power for rare variant discovery. To do this we leveraged existing whole-genome and genome-wide SNP datasets.

Population stratification might be an issue. The PCA plots suggest that not all samples cluster, but the supplementary text suggests that samples which didn’t cluster were included anyway. The authors are evidently confident that this was acceptable. More detail

about these considerations might be appropriate in the supplementary text. For example, are they satisfied that they have fully accounted for population stratification? Why do they feel that outliers improve rather than confound the analysis?

In the revised version of the manuscript, we have restricted our analysis to individuals of genetically defined European ancestry. Individuals of other ancestries have been excluded. To further account for ancestry, we have included ten principal components in the association analysis. The genomic inflation from the final GWAS meta-analysis is <1.03 indicating that population stratification is not a confounding factor.

In validation, individuals and SNPs with >50% missingness seems lenient.

This replication cohort has not been utilised in the revised version of the manuscript.

In validation, five of the association signals identified in discovery demonstrated significant evidence of association in validation (main text, line 115) – and all but the FADS2 locus remained significant after correcting for multiple testing (main text, line 116). The authors state in the supplementary text that, “The statistical significance threshold correcting for examining six independent loci was $p < 0.0083$ ” (supplementary text, line 94). Therefore, my interpretation is that four of the association signals identified in discovery demonstrated significant evidence of association in validation according to the authors own definition of significance ($p < 0.0083$). The authors subsequently discuss FADS2 as though it has been validated. If the authors prefer to define validation as $P < 0.05$, they should be clear about this.

In the revised version of the manuscript a meta-analysis approach has been employed instead of targeted variant replication. All of the now eleven loci reported achieve the genome-wide significance p-value threshold of $< 5 \times 10^{-8}$. The FADS2 locus is no longer genome-wide significant and has not been included.

The functional annotation is contemporary and informative. The luciferase reporter assay supports the functional annotation.

We appreciate these supportive comments.

Very minor comments are that the abstract cites references, which probably is not necessary, and there minor typographical errors in the supplementary text which should be corrected (e.g. lines 78 & 79).

We have addressed these comments.

Overall, I consider this an important study. It may, however, be more suitable for a specialty journal.

We respectfully disagree with the Reviewer on this point. ICP is of clinical and scientific interest to a wide range of specialists including obstetricians, hepatologists and biochemists. More broadly, whilst ICP is a relatively rare disease, affecting only 0.5-2% of

pregnancies, we argue that our study may be of interest to the wider public. Thus far, this rare disease was thought to be predominantly caused by rare highly penetrant mutations. Our study, however, demonstrates that the genetic architecture of ICP is more complex and that common sequence variants, both coding and noncoding, contribute to ICP genetic susceptibility.

There are between 5,000 and 8,000 known rare diseases, which altogether affect about 1 in 17 people at some point in their lives (PMID 31527858). However, we are still far from fully capturing their genetic basis with current approaches, which focus on coding rare mutations. Our manuscript is a strong case study in favour of considering both rare and common genetic variants to fully understand the genetics of rare human diseases, a point that will be pertinent to numerous other diseases and traits. It is therefore our opinion that this paper will be of interest to a wide readership and is appropriate for publication in a more general journal.

Reviewers' Comments:

Reviewer #2:

Remarks to the Author:

Thank you for the chance to re-review the Dixon et al manuscript. I find this version much improved and believe that if published it be a valuable addition to the body of literature investigating the genetic susceptibility to intrahepatic cholestasis of pregnancy. The caveat remains that the sample size is relatively small, but perhaps this paper can be a driver for a larger set of collaborative studies in the future as there are many small ICP cohorts out there.

Overall the GWAS methodology is sound (but see my comment for line 310 below), and there has been careful investigation of the potential impacts of associated variants on possible candidate genes. Apart from line 310 have only a few minor points that are more to do with the writing/expression of concepts rather than any methodological issues.

Line 97 and 101 – for clarity re-state the cohorts, rather than 'former' and 'latter'.

Line 144 – the ORs are really very large for a GWAS, and might come down in larger studies, but are plausible given the likely liver-specificity of genetic effects contributing to ICP.

Line 175 – write out NAFLD in full.

Line 180 – Perhaps something like 'at the remaining loci'. Writing 'in contrast' seems to set up the expectation the contributors to ICP susceptibility are expected to be coding variants, when there really should be no such expectation. Your following sentence also seems to suggest this, but in fact most GWAS hits are non-coding.

Line 218 – could remove the word 'data' or 'datasets'

Line 219 – 'encodes', rather than 'encodes for'

Line 226 – maybe write 'at this locus', rather than 'the locus'.

Line 308 – Do you mean six of your ICP loci are/were genome-wide significant in the cholelithiasis GWAS?

Line 310 – This is not an appropriate test and therefore slightly misleading, especially given the sample size of the cholelithiasis GWAS and the resultant power to detect significant signal. Better to just report the cholelithiasis P-values than to try to shoehorn significant signal with regard to ICP loci here.

Line 314 – stick with MAF rather than AF as you've previously used this abbreviation.

Line 335 – GD? Is this sentence relevant for this version of the manuscript? Perhaps wrap this sentence into the above paragraph if so?

Line 339. Split the last sentence into two. Would be slightly more impactful.

Reviewer #3:

Remarks to the Author:

The study is much better designed & powered to identify common variants associated with ICP than before. The approach to functional fine mapping & functional annotation is up to date and provides meaningful insights.

Major comments

None

Minor comments

It was not immediately obvious to me why the investigators chose to do a meta-analysis of the NIHR-RD and 100KGP panels, followed by a meta-analysis of those results with the FinnGen panel. Why not a meta-analysis of all three panels?

Page 4, lines 98 – 101: “For women that used hormonal contraception, 15/254 (5.9%) reported itching when used and 39/256 reported cyclical itching. Additionally, 15/247 patients (6.1%) reported drug-induced itching (other than for contraception)”. Personally, I found this additional information to be a distraction.

Page 4, lines 107 & 108: The source of the control data is not immediately clear in the main text (although it is clear in the Supplementary data). Could this be stated more clearly in the main text? A single sentence would be adequate.

Page 5, line 125: Release 4 of FinnGen consists of data about 176,899 individuals but 740 cases and 99,621 controls were included in the analysis. A (very) brief explanation in the main text for the reduction in numbers would be helpful.

A substantial part of the main text is dedicated to the functional fine mapping & functional annotation of risk loci. The approach is clearly described by Supplementary Figure 1b, but it was not immediately obvious in the main text. Please consider including Supplementary Figure 1b as a main figure (or summarise the approach to prioritization of causal variants more plainly in the main text).

The potential relationship between ICP and cholelithiasis (and the other conditions listed in Supplementary Figure 8) is interesting. Would a formal test of genetic correlation have been appropriate? Supplementary Figure 9 refers to enrichment analysis for numerous traits, but the main text and the supplementary data refer to derivation of polygenic risk scores for ICP and cholelithiasis. Is the enrichment analysis described anywhere in the main text or supplementary methods? (I could not find it.) Why is the PRS approach only applied to cholelithiasis when the enrichment analysis suggests significant enrichment in other conditions.

REVIEWERS' COMMENTS

Reviewer #2 (Remarks to the Author):

Line 97 and 101 – for clarity re-state the cohorts, rather than ‘former’ and ‘latter’.

This has been addressed.

Line 144 – the ORs are really very large for a GWAS, and might come down in larger studies, but are plausible given the likely liver-specificity of genetic effects contributing to ICP.

We agree and, as pointed out by the reviewer, our functional genomics analysis does prioritise liver-specific genes and regulatory elements, which fits the aetiology of ICP.

Line 175 – write out NAFLD in full.

This has been addressed

Line 180 – Perhaps something like ‘at the remaining loci’. Writing ‘in contrast’ seems to set up the expectation the contributors to ICP susceptibility are expected to be coding variants, when there really should be no such expectation. Your following sentence also seems to suggest this, but in fact most GWAS hits are non-coding.

This has been addressed

Line 218 – could remove the word ‘data’ or ‘datasets’

This has been addressed.

Line 219 – ‘encodes’, rather than ‘encodes for’

This has been addressed.

Line 226 – maybe write ‘at this locus’, rather than ‘the locus’.

This has been addressed.

Line 308 – Do you mean six of your ICP loci are/were genome-wide significant in the cholelithiasis GWAS?

Yes, we clarified the language here. It now reads:

“Six of the loci identified in our ICP GWAS meta-analysis achieve genome-wide significance for cholelithiasis as well ($p < 5e-8$).”

Line 310 – This is not an appropriate test and therefore slightly misleading, especially given the sample size of the cholelithiasis GWAS and the resultant power to detect significant signal. Better to just report the cholelithiasis P-values than to try to shoehorn significant signal with regard to ICP loci here.

We have changed this sentence to say “Two additional loci demonstrate nominal evidence of association”.

Line 314 – stick with MAF rather than AF as you’ve previously used this abbreviation.

This has been addressed.

Line 335 – GD? Is this sentence relevant for this version of the manuscript? Perhaps wrap this sentence into the above paragraph if so?

We have edited the text as suggested.

Line 339. Split the last sentence into two. Would be slightly more impactful.

We have edited the text as suggested.

Reviewer #3 (Remarks to the Author):

The study is much better designed & powered to identify common variants associated with ICP than before. The approach to functional fine mapping & functional annotation is up to date and provides meaningful insights.

We thank the reviewer for the positive feedback.

Minor comments

It was not immediately obvious to me why the investigators chose to do a meta-analysis of the NIHR-RD and 100KGP panels, followed by a meta-analysis of those results with the FinnGen panel. Why not a meta-analysis of all three panels?

The NIHR-RD and 100KGP cohorts have predominantly been recruited from the UK whereas FinnGen is from Finland. There are slight genetic differences between Finnish and non-Finnish Europeans (<https://www.nature.com/articles/nature19057>). For this reason, we elected to meta-analyse the UK cohorts together first and thereafter combine the results with FinnGen. We have reported the allele frequencies for the risk variants in the two cohorts separately in Table 1 and Supplementary Table 1.

Page 4, lines 98 – 101: “For women that used hormonal contraception, 15/254 (5.9%) reported itching when used and 39/256 reported cyclical itching. Additionally, 15/247 patients (6.1%) reported drug-induced itching (other than for contraception)”. Personally, I found this additional information to be a distraction.

We would prefer to retain this information as we were asked to include it by another reviewer, and it pertains to part of the ICP phenotype.

Page 4, lines 107 & 108: The source of the control data is not immediately clear in the main text (although it is clear in the Supplementary data). Could this be stated more clearly in the main text? A single sentence would be adequate.

We have clarified to the main text.

Page 5, line 125: Release 4 of FinnGen consists of data about 176,899 individuals but 740 cases and 99,621 controls were included in the analysis. A (very) brief explanation in the main text for the reduction in numbers would be helpful.

We have added this information to the main text, which now reads:

“There were 740 cases and 99,621 controls following FinnGen phenotype evaluation.”

A substantial part of the main text is dedicated to the functional fine mapping & functional annotation of risk loci. The approach is clearly described by Supplementary Figure 1b, but it was not immediately obvious in the main text. Please consider including Supplementary Figure 1b as a main figure (or summarise the approach to prioritization of causal variants more plainly in the main text).

We have converted the former Supplementary Figure 1b into a main figure (Figure 2) as suggested.

The potential relationship between ICP and cholelithiasis (and the other conditions listed in Supplementary Figure 8) is interesting. Would a formal test of genetic correlation have been appropriate?

We have performed genome-wide genetic correlation analysis of our ICP meta-analysis summary statistics against publicly available genome-wide summary statistics from thousands of traits using LD Score Regression (LDSC) implemented in CTG-VL (<http://vl.genoma.io/analyses/ldscore>). No trait shows a significant correlation with ICP after performing Bonferroni correction for the number of traits analysed. However, “Disorders of gallbladder, biliary tract and pancreas” does show a

nominally significant correlation ($p=0.007948$, $r_g=0.303$). Unfortunately, most of the other traits of relevance (e.g. LDL cholesterol levels, total cholesterol levels, alkaline phosphatase levels) are not present in the dataset for comparison and therefore we do not feel that this analysis adds significantly to our paper. We have thus chosen not to include it.

Supplementary Figure 9 refers to enrichment analysis for numerous traits, but the main text and the supplementary data refer to derivation of polygenic risk scores for ICP and cholelithiasis. Is the enrichment analysis described anywhere in the main text or supplementary methods? (I could not find it.)

*We thank the Reviewer for pointing this out. In the previous version of the manuscript, the enrichment analyses were described in detail in the "Online Methods" document. We have now reformatted the manuscript to contain a **Methods** and a **Supplementary Note on Methods**. The description of the enrichment analyses can be found in the section "Unbiased gene prioritisation and expression/pathway/GWAS Catalog analyses".*

Why is the PRS approach only applied to cholelithiasis when the enrichment analysis suggests significant enrichment in other conditions.

We decided to only focus the PRS approach on cholelithiasis given the much stronger enrichment of ICP-associated genes with cholelithiasis compared to other diseases/traits (Supplementary Figure 9). We have now emphasised this in the main text. Furthermore, the other traits showing significant overlap are predominantly quantitative endophenotypes (e.g., LDL cholesterol) whilst cholelithiasis, like ICP, is a binary disease state.